# Improvement Effect of Metformin on Female and Male Reproduction in Endocrine Pathologies and Its Mechanisms

**DOI:** 10.3390/ph14010042

**Published:** 2021-01-08

**Authors:** Alexander O. Shpakov

**Affiliations:** I.M. Sechenov Institute of Evolutionary Physiology and Biochemistry of Russian Academy of Sciences, 194223 Saint Petersburg, Russia; alex_shpakov@list.ru; Tel.: +7-812-5523117

**Keywords:** metformin, diabetes mellitus, gestational diabetes mellitus, polycystic ovary syndrome, in vitro fertilization, ovary, testes, insulin, gonadotropin, folliculogenesis, steroidogenesis

## Abstract

Metformin (MF), a first-line drug to treat type 2 diabetes mellitus (T2DM), alone and in combination with other drugs, restores the ovarian function in women with polycystic ovary syndrome (PCOS) and improves fetal development, pregnancy outcomes and offspring health in gestational diabetes mellitus (GDM) and T2DM. MF treatment is demonstrated to improve the efficiency of in vitro fertilization and is considered a supplementary drug in assisted reproductive technologies. MF administration shows positive effect on steroidogenesis and spermatogenesis in men with metabolic disorders, thus MF treatment indicates prospective use for improvement of male reproductive functions and fertility. MF lacks teratogenic effects and has positive health effect in newborns. The review is focused on use of MF therapy for restoration of female and male reproductive functions and improvement of pregnancy outcomes in metabolic and endocrine disorders. The mechanisms of MF action are discussed, including normalization of metabolic and hormonal status in PCOS, GDM, T2DM and metabolic syndrome and restoration of functional activity and hormonal regulation of the gonadal axis.

## 1. Introduction

Metformin (1,1-dimethyl biguanide hydrochloride) (MF), an orally administered biguanide, is a first-line drug for the treatment of type 2 diabetes mellitus (T2DM). It reduces the adipose tissue mass and increases the tissue sensitivity to insulin, thereby reducing hyperglycemia, normalizing carbohydrate and lipid metabolism and preventing inflammation and oxidative stress in the tissues [1,2]. MF is also used to treat non-alcoholic fatty liver disease [3], coronary artery disease [4,5], acute kidney injury and chronic kidney disease [6], in patients with T2DM, metabolic syndrome (MetS) and obesity, and in patients without apparent symptoms of metabolic disorders [7]. There are numerous clinical and experimental studies indicating the effectiveness of MF as an anticancer drug, used to prevent the growth and metastasis in breast cancer [8,9], endometrial cancer [10,11,12], colorectal cancer [9,13], prostate cancer [14] and in a number of the other tumors [15,16].

Currently, there is a large body of evidence for the effectiveness of MF therapy in restoration of reproductive functions and fertility in women with polycystic ovary syndrome (PCOS), gestational diabetes mellitus (GDM) and T2DM, as well as to improve the effectiveness of the assisted reproductive technologies (ART), such as in vitro fertilization (IVF) and intracytoplasmic sperm injection (ICSI). Important attributes of MF use in the treatment of pregnant women with PCOS and T2DM include its lack of teratogenic effect and established positive effect on fetal development, pregnancy outcomes and newborn health. Moreover, convincing evidence has been obtained for the restorative effects of MF on steroidogenic and spermatogenic functions in men with diabetes mellitus (DM) and MetS. This review offers an overview of problems when utilizing MF therapy for the correction of reproductive dysfunctions in women and men and includes the analysis of possible mechanisms for positive effects of MF on reproduction. The review also includes only a brief description of the molecular mechanisms of MF action in target cells; these mechanisms are the focus of other review articles [17,18,19,20,21,22,23,24].

## 2. Summary of Cell Targets and Molecular Mechanisms of Action of Metformin

The signaling pathways of MF in cells of human and mammals are still not fully understood, they seem to be dependent on species and cell type, as well as doses and routes of administration, along with metabolic and hormonal status of subjects [22,24,25,26,27].

The molecule of MF, a small hydrophilic cation, is transported from the extracellular space to the cytoplasm of the target cell through organic cation transporters-1 and -2 (OCT1, OCT2), multidrug and toxin extrusion transporters (MATE), and ATM (ataxia telangiectasia mutated) transporter, and OCT1 and OCT2 are considered as the main functional units of MF transmembrane transport [28]. The transfer of MF across the placental barrier during pregnancy is largely dependent on the transporter OCT3 [29]. The ultimate intracellular target for MF is the 5′-adenosine monophosphate-activated protein kinase (AMPK), the key energy sensor of the cell, although MF does not interact directly with the enzyme [22,30,31,32]. In pathological conditions, like T2DM and MetS, the activity of AMPK is reduced. MF’s action increases the activity of AMPK, and consequently normalizes the energy metabolism of the target cell. The AMPK consists of a catalytic α-subunit and the regulatory β- and γ-subunits that form a functionally active αβγ-heterotrimeric complex, and is widely distributed in all subcellular compartments (cytoplasmic, lysosomal, mitochondrial, and nuclear). AMPK is activated by increasing levels of AMP, a positive allosteric regulator of the enzyme [31,33,34]. The interaction of AMP with the adenine nucleotides-binding sites located in the γ subunit leads to stabilization of the αβγ heterotrimeric complex and enables phosphorylation of the α-subunit by liver kinase B1 (LKB1), which leads to the increase in AMPK activity [31,32,35] (Figure 1). Activating phosphorylation of AMPK may be also mediated by Ca^2+^-calmodulin-dependent protein kinase kinase 2 (CaMKK2) [36,37] and transforming growth factor β activated kinase-1 (TAK1) [38,39,40], but LKB1 is most important for AMPK activation [31,34,41,42,43]. Allosteric binding of AMP and ADP to γ-subunit of AMPK increases the ability of LKB1 and CaMKK2 to phosphorylate AMPK α-subunit at the Thr^172^ [44,45,46]. In the lysosomes, the “non-canonical” pathway of LKB1-mediated AMPK activation is carried out through dissociation of fructose 1,6-bisphosphate from aldolase. At the lysosomal surface, free aldolase promotes the formation of a multiprotein complex, including the vacuolar H^+^-ATPase and the scaffold protein AXIN, and this complex ensures the effective binding between AMPK and LKB1, thereby activating AMPK [47,48]. A negative regulator of AMPK is the protein phosphatase 2C (PP2C), which dephosphorylates and inactivates the α-subunit of AMPK, causing the dissociation of the αβγ-heterotrimeric complex. Elevated levels of AMP lead to an inhibition of PP2C activity, which allows AMPK to remain stable in the active Thr^172^-phosphorylated state [49,50].

MF penetrates into the mitochondria through intracellular space and accumulates in them. While in the mitochondria, MF inhibits the mitochondrial ETC complex I, which leads to decrease in ATP production and increase in the [AMP]_i_/[ATP]_i_ and [ADP]_i_/[ATP]_i_ ratios [51,52,53,54]. Moreover, MF decreases the activity of the enzyme AMP-deaminase (AMPD), which converts AMP to inosine monophosphate, inducing the accumulation of AMP within the cell [55]. The MF-induced increase in the intracellular AMP level leads to the activation of AMPK as described above [41,56]. The MF effect on AMPK activity is observed at drug concentrations below 80 μM, which are achieved with oral administration of therapeutic doses of MF [57]. The MF-induced activation of AMPK results in the stimulation of energy-producing catabolic pathways that mediate the increased glucose uptake by cells, the increased expression and activity of the membrane glucose transporters, the activated metabolic processes such as glycolysis and oxidative phosphorylation, and the normalization of mitochondrial biogenesis [20,24,58,59]. The MF-induced AMPK stimulation leads to phosphorylation of types 1 and 2 acetyl-CoA carboxylases (ACC1 and ACC2), inducing an inhibition of lipogenesis and stimulation of the β-oxidation of free fatty acids [60,61,62] (Figure 1). The ultimate results of this metabolic cascade is the decrease of T2DM- and MetS-produced dyslipidemia, and the normalization of lipid metabolism. In addition, the AMPK activation induces a plethora of cellular events, including regulation of autophagy and apoptotic processes, a decrease in the activity of inflammatory factors, including nuclear factor κB (NF-κB) and interleukin 1β, an inhibition of the ROS production, a decrease in the ER stress, as well as a decrease in insulin/IGF-1-induced activation of the mTORC1/2 complexes and a decrease in the protein synthesis [60,62,63,64,65,66].

The MF is a functional antagonist of cAMP-dependent signaling cascades, which are stimulated by hormones, glucagon in particular, through the G_s_ protein-coupled receptors and the membrane-bound forms of adenylyl cyclase (AC) [67,68]. The stimulation of AC results in an increase in the intracellular cAMP level and the activation of the protein kinase A (PKA) and the cAMP-activated transcription factor CREB (cAMP response element-binding protein). The MF-induced activation of AMPK promotes phosphorylation and activation of cAMP-specific 3′,5′-cyclic phosphodiesterase 4B (PDE4B), thereby reducing the intracellular level of cAMP [68]. Moreover, MF causes an increase in the intracellular level of AMP, a negative regulator of the catalytic site of AC, which leads to inhibition of AC activity and a decrease in cAMP production. An increase in the level of AMP can be the result of both inhibition of the mitochondrial ETC complex I, and suppression of the activity of AMP deaminase [55,69] (Figure 1). A decrease in the activity of cAMP-dependent pathways in the liver, like activation of AMPK, leads to the inhibition of glucose synthesis in hepatocytes. Furthermore, MF-induced AMPK activation induces the protein kinase ι/λ-mediated phosphorylation of cyclic AMP response element binding (CREB)-binding protein (CBP or CREBBP) at the Ser^436^, which leads to the inability of the phospho-CBP to form a functionally active complex with the factor CREB and thereby inhibits the cAMP-dependent gene transcription [70].

Along with AMPK-dependent, there are also AMPK-independent pathways of MF action on the intracellular effector systems and gene expression. High-dose MF inhibits the activity of the mitochondrial glycerol-3-phosphate dehydrogenase (mG3PDH) [71]. The inhibition of mG3PDH leads to an increase in NADH levels and decreases NAD^+^ levels, and this causes a deficiency in NAD^+^, which is involved in the conversion of lactate to pyruvate (Figure 1). Since a decrease in mG3PDH activity inhibits the conversion of lactate to glucose, the result of impaired gluconeogenesis in hepatocytes is an accumulation of lactate, which can cause lactic acidosis in the conditions of high-dose MF treatment [71,72]. Another target of MF is the enzyme H3K27me3-demethylase KDM6A/UTX, which is responsible for the transcriptional activity of a large number of genes [73].

The antidiabetic effects of MF may be due to the changes in the gut microbiota, due to stimulation of the growth of bacteria that produce short-chain fatty acids [74]. By modulating the composition of the microbiota in rodents with T2DM and MetS, MF reduces the levels of bacterial lipopolysaccharides in the blood [75], and activates AMPK-dependent pathways in the mucosal layer of the intestine, reducing glucose absorption [76].

The most important mechanism of action of MF on target cells is the enhancement of the insulin signaling pathways and the decrease in insulin resistance (IR). This may be due to inhibition of hyperactivated nuclear factor κB (NF-κB), a transcription factor that provokes the development of IR, as well as a decrease in the expression of the phosphatase and tensin homolog (PTEN), which dephosphorylates phosphatidylinositol-3,4,5-triphosphate and thereby prevents insulin-induced stimulation of Akt kinase, a key effector component in the 3-phosphoinositide signaling pathway. The inhibitory effect of MF on the activity of NF-κB-dependent signaling pathways is carried out mainly through the stimulation of AMPK [25,77,78]. Since NF-κB plays a key role in inflammatory reactions, its inhibition by MF promotes the weakening of inflammation and increases the cell survival, and these effects of MF are prevented by AMPK inhibitors [25,79,80].

## 3. Metformin and Polycystic Ovary Syndrome

### 3.1. Pathophysiology of Polycystic Ovary Syndrome

The PCOS occurs in average from 9% to 18% of women of reproductive age and includes a number of metabolic and endocrine dysfunctions [81]. Some of them are: (i) the ovarian dysfunction, characterized by irregular or no ovulation (oligo- or amenorrhea), the increased secretion of androgens (hyperandrogenism, HA) and estrogens, the endometrial hyperplasia and the increased size of the ovaries, (ii) the pancreatic dysfunction leading to insulin hypersecretion and, as a result, to insulin resistance (IR) development, (iii) the adrenal dysfunction, which leads to hyperproduction of androgens, and (iv) the functional changes in the hypothalamic and pituitary links of the female hypothalamic-pituitary-gonadal (HPG) axis [82,83,84,85]. Since these dysfunctions and changes are usually associated with obesity, MetS and T2DM, the PCOS is much more common in women with these metabolic disorders (on average in 30% of cases), with a significant proportion of PCOS patients having IR with accompanying compensatory hyperinsulinemia [86,87,88,89,90,91]. According to the Rotterdam criteria (2003), the main diagnostic criteria for PCOS are clinical or biochemical HA, oligo- or amenorrhea associated with chronic anovulation, and morphological features of PCOS, which include 12 or more follicles (2 to 9 mm) in each ovary and/or an increase in ovarian volume over 10 mL [92,93,94]. It should be noted that about 80% of women with anovulatory infertility have typical signs of PCOS [81].

The etiology and clinical manifestations of PCOS depend on many factors, as well as combinations and interactions between them. The genetic predisposition [95,96,97,98] and epigenetic factors, including an increased level of gene methylation, histone modification, and microRNA pattern variation [99,100,101], are important for the development of PCOS. Environmental and socioeconomic factors are also of great importance, including ethnic characteristics, nutrition, and adverse environmental factors (toxins, xenobiotics, chemical mutagens, and ionizing radiation) [102,103,104]. The development of PCOS in women largely depends on the effects of maternal hormones during the prenatal period, as well as on their metabolic and hormonal status in the early childhood [84,99,105,106,107].

### 3.2. The Use of Metformin in PCOS Women

In recent years, MF therapy has become widely used for correction of the metabolic and hormonal impairments in women with PCOS and for restoration of their reproductive functions [85,108,109,110], including the improvement of IVF/ICSI outcomes in PCOS [111,112,113,114]. MF is most effective in treating PCOS patients with the metabolic disorders such as T2DM, obesity, dyslipidemia, and severe IR [85,115,116]. This is majorly attributed to the alleviation of negative effects of these disorders on the female reproduction by MF, increased tissues sensitivity to insulin, improved lipid and glucose metabolism and cell metabolism, and reduced inflammation and oxidative stress in the ovaries as well as in other tissues. In cases where significant metabolic changes in PCOS patients are not observed during treatment, MF therapy can lead to energy and hormonal imbalance. The outcomes may be the opposite of improvement, but a further deterioration in reproductive functions. This possibility is supported by the data from clinical trials on metabolic changes, including an increase in fasting glucose clearance and endogenous glucose production [117,118], as well as changes in the microbiota in non-diabetic individuals [119], as well as data on metabolic and hormonal dysfunctions in normal rodents, for a long time receiving MF [120].

There is a lot of clinical evidence of the high efficacy of MF in PCOS, which makes it feasible to consider MF as a second-line drug for ovulation induction in women with PCOS [109,121,122,123,124,125,126,127,128]. MF is recommended for the induction of ovulation in PCOS women who are either resistant to clomiphene citrate (CC) or require antiandrogen therapy without the use of contraceptives [125], as well as in PCOS patients with severe obesity and impaired lipid metabolism [114]. One very important consideration during PCOS treatment with MF is that drug has no or little adverse effects on the outcomes of pregnancy as well as the health of fetus and newborn, which indicates the safety of MF therapy [126,127]. The gastrointestinal side effects of MF have been reported in a number of cases, but these effects did not significantly affect the health of PCOS women [108,112].

The MF treatment of PCOS women normalizes the frequency and regularity of ovulation, including when co-administered with exogenous gonadotropins [112,129,130]. This suggests that MF can also affect the sensitivity of ovarian cells to gonadotropins, which is important for the ART. As a result, during the ART, the most promising approach is the combined use of MF with gonadotropins [111,113]. In PCOS, MF improves clinical pregnancy rates and live birth rates [108,111,112,113,131,132,133,134,135,136], and also reduces the number of miscarriages and increases the rate of embryo implantation [137,138].

There is evidence of a positive effect of MF on the effectiveness of IVF and IVF/ICSI in PCOS women [114]. It is believed to be due to the normalization of metabolic and hormonal parameters and the androgen levels in PCOS, which leads to an improvement of embryo implantation, an increase in the ovarian response to gonadotropins and a decrease in the rates of miscarriage [112,135,138,139,140,141]. The increased gonadotropin sensitivity allows avoiding the use of high-dose gonadotropins and, thereby, preventing the ovarian hyperstimulation syndrome (OHSS), a severe complication of gonadotropin-induced ovulation induction. However, it should be noted that some data on the use of MF in the ART technology in PCOS women are not so unambiguous, and there are results that do not support the efficacy of MF in IVF/ICSI. The clinical studies carried out by Egyptian group of physicians showed no improvement in IVF rates in PCOS women who received MF [142]. However, in this study, overweight or obese PCOS women received short-term courses of low-dose MF (1000 mg/day), from the start of ovarian stimulation with gonadotropins until proof of clinical pregnancy. As a result, in this case, the period of time for the manifestation of the restorative effects of MF on the ovaries and folliculogenesis in PCOS patients may not have been long enough. Potentially, for an adequate estimation of MF effectiveness in PCOS patients it is necessary to separate them in the groups, based on the severity and duration of the disease and in the body mass index [108,114], as well as the severity of IR, dyslipidemia and hyperglycemia.

### 3.3. Combined Use of Metformin with Clomiphene Citrate, Letrozole, Liraglutide, Saxagliptin, or Oral Contraceptives

A promising approach to treat PCOS is the use of combination of MF with the other drugs that improve the ovarian function and metabolic parameters in PCOS, with the best candidates for co-administration are CC, a mild nonsteroidal estrogen antagonist belonging to the family of selective estrogen receptor modulators, and letrozole, a non-steroidal aromatase inhibitor that prevents the conversion of androgens to estrogens [116,131,143,144,145,146,147,148].

The CC is the main drug of choice for treatment of PCOS, yet a significant proportion of PCOS women have weak or no response to CC therapy. Therefore, a search is underway for drugs that can potentiate the therapeutic effects of CC in PCOS, and MF is one of the most promising candidates [108,131,134,143,144,146]. Combined use of MF plus CC in PCOS showed significant improvement in clinical indices of pregnancy and the combination therapy is more effective than the use of CC alone. However, a number of studies reported no effect [149] or relatively weak potentiating effect of MF for CC therapy [134]. One of the possible reasons for these contradictory results may be the difference in the sensitivity to CC and MF in PCOS patients. The most profound potentiating effect of MF on the induction of ovulation and pregnancy rates is found in patients with a pronounced resistance to CC [144,146,150]. However, some PCOS patients may be also insensitive to MF, which is due to many factors, including the polymorphisms and inactivating mutations in the transmembrane proteins facilitating intracellular transport MF [151]. As a result, the combined therapy is expected to benefit mainly PCOS patients with reduced sensitivity to CC, pronounced obesity, IR and dyslipidemia, and high sensitivity to MF.

In recent years, the data have been obtained for the effectiveness of the combined use of MF and letrozole, an aromatase inhibitor that is widely used to restore the ovarian cycle and induction of ovulation and improves oocyte implantation and pregnancy rates in women with PCOS, primarily those with reduced sensitivity to CC [144,148,150]. The combined therapy with MF demonstrated enhancement for the improving effects of letrozole on the pregnancy and live birth rates. Moreover, there are clinical results showing that the combined use of letrozole and MF is more effective than the combined use of CC and MF [145,147].

In PCOS patients, the efficiency of MF therapy is increased when MF is used with oral estrogen-progestin contraceptives, both acting similar, by suppressing ovarian androgen overproduction and normalizing menstrual cycle, most noticeably in obese PCOS women [152,153]. On a contrary, when the same combined treatment (MF and oral contraceptives) is used for PCOS women with normal or reduced body weight, it results in a decrease in their muscles mass, leads to the water retention and the formation of an “osteosarcopenic” phenotype [154]. Two main reasons are behind the decrease in the muscles mass during combined therapy. First, MF and oral contraceptives reduce the blood androgen levels. It is known that in PCOS there is a significant positive correlation between the blood level of androgens and the muscles mass [155]. Second, MF-induced activation of AMPK and changes in mitochondrial energy status stimulate catabolic processes in the muscles tissue, which leads to muscles atrophy, as shown in patients with T2DM [156]. In this regard, it should be noted that MF treatment of T2DM patients leads to an increase in the blood level of fibroblast growth factor 21 (FGF21), which is one of the specific markers of muscles damage and degeneration [157]. Thus, it is highly recommended to take into account the proportion of the muscle tissue and body mass index in PCOS women, as well as the severity of HA when considering the option of using the combined therapy of MF and oral contraceptives [154].

The agonists of glucagon-like peptide-1 (GLP-1) receptor and the inhibitors of dipeptidyl peptidase-4 are widely used to treat T2DM and MetS [158,159,160,161], but they can also be used to correct the metabolic alterations and IR in PCOS women, as well as in pregnant women with GDM and T2DM [162,163]. It is shown that MF enhances the beneficial effect of liraglutide, a selective GLP-1 receptor agonist, on insulin sensitivity and glucose homeostasis. The 12-week treatment of 30 obese PCOS women with a combination of MF (1000 mg twice a day) and liraglutide (1.2 mg/day) causes a decrease in IR and normalizes the sensitivity of patients to glucose, and the combined therapy was more effective than monotherapy [164]. The treatment of premenopausal PCOS women with MF (2000 mg/day), saxagliptin (5 mg/day), an inhibitor of dipeptidyl peptidase-4, or a combination of MF and saxagliptin leads to normalization of glucose tolerance on average of 56% of patients [165]. Moreover, in the group treated with MF alone or saxagliptin alone, the improvement of glycemic control is demonstrated only in 25 and 55% of patients, respectively, while the combined therapy restores glucose tolerance in 91% of women with PCOS [165]. A high efficacy of the combined therapy was shown by other group of authors who monitored the 16-week treatment of 38 women with pre-diabetes and PCOS using the MF plus saxagliptin [166]. Weight loss and decrease in hyperglycemia and IR, which are induced by treatment of obese PCOS patients with GLP-1 receptor agonists, lead to a decrease in HA [167,168,169] and an improvement in menstrual frequency [167,169]. Liraglutide, an analogue of GLP-1, normalizes the menstrual cycle and fertility in women with HAIR-AN syndrome, which is due to a decrease in the levels of androgens and insulin [170]. Consequently, in PCOS patients, MF-induced potentiation of the metabolic-improving effects of GLP-1 agonists may also increase their restorative effects on the menstrual cycle and fertility.

### 3.4. The Mechanisms of Metformin Effects on Reproductive Functions in PCOS

#### 3.4.1. Metformin-Induced Inhibition of Hyperandrogenism and Normalization of the Steroid Hormones Balance

One of the main mechanisms of the restorative effect of MF on ovarian function, ovulation and pregnancy in PCOS women mediates through the pronounced antiandrogenic effect of MF, both in monotherapy and in combination with other drugs [141,152,171,172,173,174,175,176,177,178,179]. One-year treatment of overweight PCOS women with MF (1700 mg/day) reduced the levels of free testosterone, dehydroepiandrosterone (DHEA) and androstenedione, and significantly weakened the signs of hirsutism. This effect of MF was strongly associated with a decrease in the homeostasis model assessment of insulin resistance index (HOMA-IR) and an improvement in glucose tolerance, but was weakly associated with a decrease in the body weight, which indicates a main contribution of a decrease in IR and hyperinsulinemia to the antiandrogenic effect of MF [180]. An antiandrogenic effect was demonstrated in the treatment of overweight and obese adolescents with MF (1000–2000 mg/day), and was accompanied by a significant decrease in IR [152,173,174,176,177]. MF reduced both the basal and gonadotropin-stimulated testosterone levels, and these effects were observed even with short-term MF treatment. The administration of MF for two days to PCOS women caused a decrease in their testosterone levels stimulated by luteinizing hormone (LH). This effect was not due to a decrease in the body weight and the changes in metabolic indices, pointing to potential direct influence of MF on steroidogenic activity in ovarian cells [181].

In PCOS, the severity of IR is positively correlated with the severity of HA and dysregulations of the ovulatory cycle. The PCOS women with oligomenorrhea and without HA usually do not have IR, while the PCOS women with oligomenorrhea and HA often show significant signs of IR [182]. In turn, in PCOS women with regular ovulatory cycle, IR was less pronounced than in women with PCOS and irregular or no ovulation [183].

The inhibitory effect of MF on the production of steroid hormones by ovarian cells was demonstrated in the in situ experiments using different cell lines [94,184,185,186]. Cultured human ovarian cells grown in the presence of MF, showed a decrease in production of basal and gonadotropin- or insulin-stimulated steroid hormones. Similar effects were shown for progesterone and estradiol in granulosa cells and androstenedione in theca cells. Inhibitory effect of MF was dose-dependent and most pronounced in the measure of suppression of hormone-stimulated steroidogenesis [185]. MF (10 mM) treatment of bovine granulosa cells isolated from small follicles led to a decrease in both the basal and follicle-stimulating hormone (FSH)- and IGF-1-stimulated production of progesterone and estradiol [186].

When deciphering the mechanisms of the inhibitory effect of MF on steroidogenic activity in the ovaries, a key role is assigned to stimulation of AMPK, and the triggering of AMPK-dependent pathways in ovarian cells [94,186,187] (Figure 2). The mechanisms of AMPK activation in ovarian cells are the same as described in the Section 2 above, and are triggered by MF-induced inhibition of electron transport chain in mitochondrial respiratory complex I [188]. It is worth noticing that in humans, other mammals and birds (cows, goats, sheep, pigs, rats, mice, chicken), AMPK is widely expressed in different types of the ovarian cells (oocyte-cumulus complexes, granulosa cells, and theca cells) and in the corpus luteum [186,187,189].

There is a large body of experimental data that AMPK is essential for the regulation of folliculogenesis and meiotic activity, both control the maturation of oocytes [186,190,191,192,193,194], and that AMPK is involved in the regulation of steroidogenesis in ovarian granulosa cells [187,195]. Deletion of the AMPK α1-subunit in mouse oocytes leads to a 27% decrease in litter size, and after IVF, the number of embryos in these mutant mice decreases by 68% [196]. In the ovaries of mutant mice, the levels of transmembrane connexin-37 and *N*-cadherin, which mediate the intercellular communication and are involved in the formation of the oocyte-cumulus complexes, were significantly reduced. The activity level within cAMP-dependent cascade, which includes PKA and factor CREB, and the activity of mitogen-activated protein kinase (MAPK) cascade are reduced, indicating weakening of cAMP- and MAPK-mediated signal transduction [195,196]. The components of these signaling pathways are involved in the junctional communication between the oocyte and the cumulus/granulosa cells. The MII oocytes in mice lacking the α1-AMPK have a significantly reduced intracellular ATP level and decreased levels of cytochrome *c* and peroxisome proliferator-activated receptor γ coactivator 1-α (PGC1α), which indicates the impaired mitochondrial biogenesis and the activation of apoptotic processes [196].

In ovarian cells, through the activation of AMPK, MF inhibits the cAMP signaling pathways, decreases the expression and activity of the steroidogenic enzymes and the production of androstenedione, a precursor of testosterone (Figure 2). When exposed to cultured human theca cells, both in the basal and forskolin-stimulated state, application of MF (50 and 200 μM) caused the AMPK stimulation and reduced androstenedione synthesis in dose-dependent manner. In theca cells stimulated by forskolin, a non-hormonal AC activator, MF suppressed the expression of *StAR* and *Cyp171a* genes encoding StAR protein, which carries out cholesterol transport into mitochondria (the first, rate-limiting stage of steroidogenesis), and cytochrome P450c17α, which catalyzes the synthesis of androstenedione [184]. The inhibitory effect of MF on steroidogenesis in the rat and bovine granulosa cells was also due to AMPK activation, as indicated by an increase of Thr^172^ phosphorylation of AMPK α-subunit, as well as an increase of Ser^79^ phosphorylation and inhibition of the main target of AMPK, the enzyme acetyl-CoA-carboxylase [186,197]. The involvement of AMPK in the antiandrogenic action of MF is supported by the data on a similar effect of 5-aminoimidazole-4-carboxamide ribonucleotide (AICAR), the pharmacological activator of AMPK [186]. The MF-induced stimulation of AMPK in granulosa cells results in a decrease in the activity of MAPK cascade, primarily the kinases ERK1/2, and inhibition of phosphorylation of MAPK-activated protein kinase-1, also referred to as ribosomal s6 kinase (p90^RSK^) [186,197,198]. Therefore, through the AMPK-dependent mechanisms, MF not only reduces the excessive steroidogenic activity in ovarian cells, but also suppresses and/or modulates the cell growth and intracellular protein synthesis.

An important role of AMPK-dependent mechanisms in the antiandrogenic action of MF in PCOS is supported by the data on the relationship between the androgen production and the activity of LKB1 in the ovaries of mice with experimental HA [199]. The LKB1 expression in the ovaries of hyperandrogenic mice is inhibited by high concentrations of androgens through activation of intracellular androgen receptors. In opposite, LKB1 activation leads to a decrease in the androgens production by theca cells, but increases the estrogen production by granulosa cells. Transgenic mice overexpressing LKB1 are characterized by the increased resistance to the development of HA [199]. Mice with a functionally inactive ovarian gene *Lkb1* have significantly enlarged ovaries and activated entire pool of primordial follicles, but without further maturation and ovulation, which results in the premature ovarian failure and severely reduced fertility [200]. The data indicates that MF-induced activation of LKB1/AMPK pathway in PCOS ovaries normalizes ovarian steroidogenesis and counteracts HA (Figure 2).

Another, very important mechanism of antiandrogenic action of MF is largely due to an MF-induced increase in insulin sensitivity and consequent weakening of compensatory hyperinsulinemia, the main pathogenic factor in PCOS, also closely associated with HA [201,202,203] (Figure 2).

It is generally accepted that the stimulating effect of hyperinsulinemia on the production of androgens by ovarian cells is based on low affinity binding of insulin with IGF-1 receptors [203,204,205]. In the 1990s, it was shown that insulin in vitro and in vivo activates the IGF-1 receptor in the ovaries, which leads to an increase in the synthesis and secretion of androgens. This is supported by the data on the increased secretion of androstenedione and the elevated basal and LH-stimulated production of testosterone in cultured ovarian theca and stroma cells incubated in the presence of insulin [204,206]. A decrease in insulin secretion induced by MF (500 mg three times daily) in obese PCOS women led to inhibition of cytochrome P450c17α activity in the ovaries, decreasing the basal levels of 17α-hydroxyprogesterone and the levels of this hormone stimulated by leuprolide, a gonadotropin-releasing hormone (GnRH) analogue [171].

Along with the activation of IGF-1 receptor, hyperinsulinemia reduces the production of insulin-like growth factor-binding protein-1 (IGFBP-1) by ovarian granulosa cells [204] (Figure 2). This protein specifically binds IGF-1, decreasing the concentration of free IGF-1 and weakening its stimulating effect on the IGF-1 receptor and steroidogenesis in the ovarian theca and stromal cells. The result of insulin-induced decrease in IGFBP-1 level is overproduction of androgens and the impaired folliculogenesis and ovulatory cycle [207]. It should be noted that IGF-1, like insulin, reduces the IGFBP-1 production in ovarian cells, while FSH stimulates the IGFBP-1 production, preventing IGF-1-induced stimulation of androgen production [207]. The blood IGFBP-1 levels in PCOS women are significantly lower compare to than in healthy women, which may indicate suppression of IGFBP-1 production under the conditions of hyperinsulinemia [201,203]. At the same time, there is no strong correlation between the IGFBP-1 deficiency and the severity of hyperinsulinemia and IR, which suggests the presence of additional mechanisms mediating the inhibition of IGFBP-1 production in PCOS [203]. There are no data for the MF effect on blood IGFBP-1 levels in PCOS, but there is evidence of its significant increase in MF-treated women with GDM [208]. MF caused an increase in both phosphorylated and non-phosphorylated forms of IGFBP-1, thereby reducing the negative effect of IGFBP-1 deficiency on the course and outcomes of pregnancy [208].

In PCOS, the blood levels of androgen and sex hormone-binding globulin (SHBG) are reduced, which leads to an increase in free testosterone level and free androgen index. As early as the 1990s, hyperinsulinemia was found to be an important factor for the suppression of SHBG production in PCOS [171,209,210]. Overweight plays a key role in this process, as supported by observation that weight loss in obese PCOS women induced by a low-calorie diet leads to a restoration of SHBG levels, which may be due to a decrease in IR and insulin levels [211]. MF administration increases the production of SHBG, by reducing body weight and hyperinsulinemia, and, thereby, reduces the signs of HA in PCOS women [171,212,213] (Figure 2). Blood SHBG levels are also increased in MF-treated obese women without clear signs of PCOS [214]. The PCOS women with low SHBG levels are more sensitive to MF therapy, while the effectiveness of MF in patients with normal or high SHBG levels and, as a consequence, without signs of HA is significantly less noticeable. The PCOS women with an average SHBG level in the blood of 37.5 nmol/L respond well to MF treatment, while in PCOS women with an average SHBG level of 56.0 nmol/L, the response to MF was weak [215]. Therefore, the assessment of the blood SHBG concentration is an important prognostic factor for predicting the effectiveness of MF therapy in PCOS. Thus, a MF-induced decrease in the level of insulin should lead to a weakening of the stimulating effect of insulin on the ovarian IGF-1 receptors, prevent their stimulation by an excess of free IGF-1, and reduce the blood level of free androgens by restoring the SHBG production.

Of great importance for normalization of the steroidogenic function in the ovaries can be MF-induced decrease in the blood LH level and the LH/FSH ratio, which are significantly increased in PCOS [216,217,218,219] (Figure 2). An increase in the LH/FSH ratio due to abnormal gonadotropin pulsatility and hypersecretion of LH by the pituitary is a significant factor responsible for the deterioration of folliculogenesis and oogenesis in PCOS [219,220,221,222,223,224]. In most cases, gonadotropin imbalance is found in PCOS women with obesity, and level of increase in LH is correlated with the severity of obesity [220]. The restoration of this ratio leads to normalization of the ovulatory cycle and triggers the development of the dominant follicle, improving the rate and outcomes of pregnancy [222,225]. Eight-week treatment of PCOS women with MF (1500 mg/daily) results in a 32% decrease in the blood LH levels and a 42% decrease in the LH/FSH ratio [218]. There is reason to believe that, as in the case of SHBG, the sensitivity of PCOS women to MF therapy depends on the LH level and the LH/FSH ratio. The MF treatment of PCOS women with severely impaired gonadotropin secretion and significantly increased LH level is more effective than the same treatment of PCOS women without gonadotropin imbalance [226].

It is suggested that the restoration of normal LH secretion by the pituitary gland may be due to MF-induced normalization of AMPK-dependent signaling in hypothalamic neurons secreting GnRH [227] (Figure 2). This is due to the ability of MF to cross the blood-brain barrier and reach the hypothalamus and the other brain regions [228]. The secretion of GnRH is under the control of neuropeptides, such as kisspeptin, melanocortins, agouti-related peptide and neuropeptide Y, as well as γ-aminobutyric acid and the other biogenic amines [229], and the neurons producing these neurohormones may also be pharmacological targets for MF. The restoration of functional interaction between GnRH-expressing neurons and the other components of the neuronal network responsible for hypothalamic control of the HPG axis can prevent PCOS-associated HA and provide a balance between steroid hormones, thereby normalizing the functionality of feedback loops in this axis. Our group and other authors have demonstrated the restoring effect of MF on leptin signaling pathways in the hypothalamus of animals with metabolic disorders and IR [230,231,232,233], and the improvement of hypothalamic leptin signaling can also make a significant contribution to the restoration the reproductive functions in PCOS. It should be noted that MF, both at the periphery and in the CNS, acts on the signaling and effector systems synergistically with leptin. It is generally accepted that leptin is the most important regulator of the female and male reproductive systems, which stimulates the activity of hypothalamic GnRH-expressing neurons, and affects the other links of the HPG axis [234,235].

#### 3.4.2. Protective efFect of Metformin against Excess Androgens in PCOS

In addition to reducing HA in PCOS, MF is able to prevent the negative effect of excess androgens on the ovarian cells [236,237]. It was shown in mice model of PCOS, that treatment using MF (500 mg/kg, 20 days) after DHEA induction improved the quality of oocytes and normalized the early stages of embryonic development. In the ovaries of MF-treated mice, the restoration of the number of metaphase II oocytes, mitochondrial membrane potential and ATP levels were shown. Along with this, MF attenuated oxidative stress, as indicated by a decrease in reactive oxygen species levels and an increase in the reduced form of glutathione [236].

The inhibition of endoplasmic reticulum (ER) stress and the prevention of MAPK cascade hyperactivation make a significant contribution to protective effects of MF in PCOS-associated HA. The activation of ER stress in the ovaries and the triggering of signaling pathways induced by the unfolded protein response lead to impaired synthesis and post-translational modification of proteins and the mitochondrial dysfunction, all of which negatively affects folliculogenesis and meiotic maturation of oocytes [238,239,240]. The effector components of MAPK cascade p38-MAPK is important for the activation of the unfolded protein response signaling and apoptosis in ovarian cells [241,242]. More recently, it was shown that an excess of androgens led to activation of ER stress and apoptosis in human and mouse cumulus cells [237,243]. The MF treatment reduces ER stress and inhibits p38-MAPK phosphorylation, which is significantly increased in cumulus cells of PCOS women and in the granulosa cells and the oocyte-cumulus complexes in mice with DHEA-induced PCOS [237]. There is every reason to believe that this effect of MF is based on its ability to inhibit HA.

#### 3.4.3. Effects of Metformin on FSH-Activated Signaling in the PCOS Ovaries

Another mechanism of the MF restoring effect on ovarian function in PCOS is the inhibition of expression of the *Cyp19a1* gene encoding aromatase. Reduced levels of aromatase result in a decrease in estrogen response to FSH, insulin and IGF-1 in the ovaries [185,244,245,246]. A large number of PCOS patients have increased sensitivity of granulosa cells to stimulation with FSH, insulin, or IGF-1. This is due to the fact that in granulosa cells of PCOS women, the expression of the FSH and IGF-1 receptors and the IRS1 and IRS2 proteins are significantly increased [247,248,249,250,251,252]. In addition, the expression of PTEN, a negative regulator of signaling pathway involving insulin/IGF-1 receptors, IRS proteins, phosphatidylinositol 3-kinase (PI 3-K) and Akt-kinase, is reduced, which leads to hyperactivation of Akt-kinase by insulin and IGF-1 [251]. In the PCOS ovaries, the important mechanism of suppression of PTEN expression and hyperactivation of the insulin and IGF-1 signaling pathways is an increase in the expression of two microRNAs of miR-200 family, miR-200b and miR-200c, which negatively affect the expression of the *PTEN* gene [252]. In addition, a decrease in the expression of miR-99a, a negative regulator of IGF-1 receptor expression, leads to an increase in the sensitivity of granulosa cells to IGF-1 [218]. An increase in the expression and activity of the receptor and postreceptor components of the FSH-, insulin- and IGF-1-regulated signaling systems in PCOS results in the accelerated growth and proliferation of the ovarian cells, primarily granulosa cells, in the response to the stimulating effect of these hormones. Moreover, this potentiates already pre-existing increased ovarian reactivity and premature luteinization [207,253,254].

MF reduces the expression of FSH receptors thereby weakening the stimulating effects of FSH on steroidogenesis and proliferation of granulosa cells, increased in PCOS, which leads to the normalization of folliculogenesis and ovulation. Under the conditions of ovarian dysfunctions in PCOS, MF treatment postpones the triggering of processes that ensure the normal growth of antral follicles, thus providing more appropriate window of time required for their differentiation and development (on average about three months) [245]. By reducing the ovarian sensitivity to FSH, MF prevents the OHSS, the most common complication of gonadotropin-stimulated induction of ovulation [135,136,138,255,256].

The inhibitory and modulating effects of MF on the effector components of gonadotropin-stimulated cascades in ovarian cells can be realized through both AMPK-dependent and AMPK-independent pathways, including the MAPK cascade [244,245]. Through AMPK-independent pathways, MF reduces FSH-induced increases in aromatase activity and estradiol synthesis in granulosa cells, and this effect is not reproduced when using AICAR [245]. The inhibitory effect of MF on the expression and activity of aromatase can be elicited through at least three well understood mechanisms.

The first mechanism is MF-induced inhibition of the expression of FSH receptor in granulosa cells, which reduces the stimulatory effect of FSH on the intracellular signaling pathways through which FSH controls the expression of aromatase and steroidogenic enzymes [245]. As noted above, in granulosa cells of women with PCOS, the expression of the *Fshr* gene is often significantly increased, which causes the elevated responsiveness of the ovaries to FSH [247,248,249]. The polymorphisms in the *Fshr* gene can have a significant role in modulating the responsiveness to FSH in both, positive and negative way, although in PCOS the data on the interrelation between *Fshr* isoforms and the activity of FSH receptor are contradictory [257]. At the same time, there is evidence that some polymorphisms can lead to an increase in the sensitivity of FSH receptor to gonadotropin [258,259]. The second mechanism of the inhibitory effect of MF on aromatase activity is due to a decrease in FSH-induced phosphorylation of the transcription factor CREB, which positively regulates the expression of the aromatase gene, as well as the *Star*, *CYP11a1* and *HSD3b* genes encoding the cholesterol-transporting protein StAR, cytochrome P450scc (CYP11A1), which catalyzes the synthesis of pregnenolone, and 3β-hydroxysteroid dehydrogenase (3β-HSD), which converts pregnenolone to progesterone [245]. The third mechanism involves inhibition of FSH-induced dephosphorylation of the CREB-regulated transcription coactivator 2 (CRTC2) and its translocation into the nucleus, where CRTC2 is involved in the assembly of CREB containing activating transcriptional complex [245]. Thus, MF inhibits the formation of the CREB-CBP-CRTC2 activation complex, which is capable of binding to the CRE regulatory elements in the promoter of the genes encoding aromatase and some steroidogenic proteins, and prevents their overexpression by FSH.

The FSH- and insulin/IGF-1-activated signaling pathways in ovarian cells are closely interrelated due to cross-talk between them, including their interaction through the PKA/PI 3-K/Akt pathway [260,261]. The functional activity of Akt kinase is increased as a result of these pathways activation in the conditions of overstimulation of FSH receptor and FSH-mediated activation of IRS1 protein. This leads to an increase in the survival of ovarian cells, suppresses atresia of the follicles and impairs maturation of the dominant follicle.

The FSH-dependent pathways are also regulated by the members of the transforming growth factor β (TGF-β) family, including activins, inhibins, and anti-Müllerian hormone (AMH). Activins increase the FSH receptor expression and enhance the stimulatory effects of FSH on ovarian steroidogenesis [262,263]. In contrast, AMH and inhibins suppress the stimulatory effects of FSH on folliculogenesis [249,262,264,265]. The regulatory effects of the protein members of the TGF-β-family on the FSH-dependent signaling pathways can be realized through both Akt- and cAMP-dependent mechanisms [266,267,268,269], but the effect of MF on the TGF-β-mediated regulation of the FSH signaling pathways in granulosa cells remains poorly understood.

#### 3.4.4. The Effect of Metformin on the Production of Anti-Müllerian Hormone in PCOS

In PCOS, one of the targets of MF therapy is AMH, a dimeric glycoprotein that is produced by the granulosa cells of the primary, preantral and small antral follicles [270,271]. AMH concentration in the blood of women positively correlates with the follicular reserve and, as a result, in PCOS, the blood levels of AMH are usually increased by two or more fold [251,271,272,273,274]. Excess levels of AMH lead to an impaired folliculogenesis, preventing the recruitment of primordial follicles into the pool of growing follicles and reducing the responsiveness of growing follicles to FSH [249,275,276,277]. The increased levels of AMH may be due to HA and hyperinsulinemia, which are characteristic features of PCOS and are closely interrelated [249,273,278,279,280], as well as to an increase in blood LH levels or the sensitivity of granulosa cells to LH, typical for PCOS patients [280,281,282,283]. In in vitro experiments using lutein granulosa cells obtained from oligo/anovulatory PCOS women, the LH increases the AMH production, while the expression of type II AMH receptors in these cells does not change significantly. In the case of lutein granulosa cells obtained from healthy women and normo-ovulatory PCOS women, the stimulating effect of LH on AMH production is almost completely inhibited, but the inhibiting effect of LH on the expression of type II AMH receptors is preserved. This effect was reproduced with the use of cAMP analogs, which indicates the participation of cAMP-dependent mechanisms in it [281]. All this indicates that PCOS women with an impaired ovulatory cycle have an increase in both the LH-induced AMH production and the responsiveness of lutein granulosa cells to this factor.

By lowering insulin and androgen levels and normalizing gonadotropin levels, MF attenuates ovarian AMH secretion, which leads to a decrease of its inhibitory effect on folliculogenesis and a weakening of the signs of PCOS [270,271,284,285,286,287,288]. Eight-week treatment of PCOS women with MF (1500 mg/day) reduced the blood AMH levels from 10 ± 3.75 to 7.8 ± 3.7 ng/mL [271]. Six-month treatment with MF at the same dose led to a decrease in AMH, ovarian volume and antral follicle number in PCOS women [270]. The treatment of PCOS women with MF at the doses of 850 mg/day (first week), 850 mg/12 h (second week) and 850 mg/8 h (next six weeks), along with the restoration of ovulation and the normalization of LH and testosterone levels, caused a decrease in the blood AMH levels, from 8.99 ± 0.99 to 6.28 ± 0.46 ng/mL [286]. The combined therapy with MF and resveratrol of rats with DHEA-induced PCOS reduced ovarian size, improved ovarian follicular cell architecture, and decreased AMH production [217]. It is assumed that in PCOS, a decrease in the blood AMH level to control values can be considered as one of the prognostic factors of the effectiveness of MF therapy [285,286,289].

However, there are clinical studies that showed a weak suppressive effect of MF therapy on AMH production in PCOS [290], or the absence of this effect [291,292]. This may be due to differences in the MF doses, the duration of MF treatment, and the peculiarities of the hormonal status in PCOS patients. MF treatment of PCOS women for 8 months led to a decrease in the blood AMH level, while four-month MF therapy had a little effect on AMH concentration, although it significantly reduced the blood level of androstenedione and normalized the regularity of the menstrual cycle [284]. It is possible, that normalization of the AMH level at the first stage requires normalization of androgens, insulin and gonadotropins levels, which in turn affects the expression and secretion of AMH. On the other hand, Iraqi scientists showed that the treatment of PCOS women with MF (500 mg three times daily) for three or six months significantly reduced the blood AMH levels, but in the case of six-month therapy, the MF effect became less pronounced [289]. There is reason to believe that the severity of obesity, as well as the degree of an increase in AMH levels, can have a significant effect on the inhibitory effect of MF on ovarian AMH production. The most pronounced inhibitory effect of MF on AMH production was demonstrated in PCOS women with a higher body mass index, as well as with a higher level of AMH [285].

Based on the above results, as well as on the available data on the molecular mechanisms mediating the regulation of AMH production by granulosa cells [249,293], it can be assumed that the main mechanism for the improving effect of MF on AMH levels in PCOS is the weakening of HA. Under normal conditions, the androgens produced by theca cells induce a decrease in AMH levels, which leads to inhibition of the antral follicle development and precedes ovulation. With prolonged exposure to high concentrations of androgens, which are comparable to those in PCOS, the response of granulosa cells to androgens is impaired, resulting in the absence of an androgen-induced fall in AMH levels and dysregulation of follicular development [293]. When PCOS patients are treated with MF, their androgen levels are normalized, and hyperinsulinemia, which is usually associated with HA, is reduced, which leads to the restoration of the granulosa cell response to androgens. Accordingly, the low efficacy of MF in reducing AMH levels and restoring follicular maturation in patients with PCOS may be due to initially mild HA and IR. It is impossible to exclude the direct effects of MF on the production of AMH by follicular cells, including through AMPK-dependent mechanisms, as well as through weakening the stress of the endoplasmic reticulum, stimulated by high concentrations of androgens [237]. However, this issue has not yet been studied.

There are only two clinical studies on the MF effect on the levels of other TGF-β family factors. One of them showed the normalization of the TGF-β level in the blood of PCOS women with MF and cyproterone acetate/ethinyloestradiol treatment [294]. The other authors demonstrated that a long-term therapy with MF (more than three months) led to a significant decrease in the blood level of inhibin-B in PCOS patients [289].

#### 3.4.5. Effect of Metformin on Metalloproteinases in PCOS

Women with PCOS usually have the increased serum levels of type 9 matrix metalloproteinase (MMP-9) and MMP-2 [295,296], and the altered concentrations of the types 1 and 2 tissue inhibitor of MMP (TIMP-1 and TIMP-2) [295]. The changes in the concentrations and balance of MMPs and TIMPs lead to the remodeling in the ovarian stroma, increased ovarian angiogenesis and impaired folliculogenesis [295,296]. The activation of AMPK causes a decrease in the activity of the pathway involving mTOR (mammalian target of rapamycin) and ribosomal protein kinase p70S6K, and suppresses the expression and functional activity of MMPs. Consistently, stimulation of Akt kinase and mTOR enhances the expression of MMPs and, thereby, stimulates angiogenesis, cell migration, cell proliferation and protein synthesis [297,298].

By activating AMPK, MF reduces the expression and activity of MMPs, and this is largely responsible for the well-described antitumor effect of MF [299]. MF treatment (10 mM) of cultured human ovarian granulosa (HTOG) cells, leads to an increase in AMPK activity, an inhibition of mTOR-dependent cascade, and a decrease in the expression of MMP-2 and MMP-9 [300]. Another MF-mediated mechanism for down regulating the MMP expression involves triggering the H19/miR-29b-3p signaling pathway. MF increases the expression of miR-29b-3p, a negative regulator of the MMP-2 and MMP-9 expression via decreasing the expression of histone H19 through methylation of the promoter in the gene encoding this histone, which leads to a decrease in the production of these enzymes and contributes to normalization of ovarian morphology in PCOS [300]. MF-induced activation of the H19/miR-29b-3p pathway is attenuated by the increased expression of histone H19, which decreases the expression of miR-29b-3p. The miR-29b-3p-mediated effect of MF on MMP expression is independent of AMPK, in contrast to MF regulation of mTOR signaling cascade [300].

#### 3.4.6. Influence of Metformin on Inflammation and Lipid Status in PCOS

Dyslipidemia, oxidative stress and inflammation, along with IR, are essential for the pathogenesis of PCOS. These factors are similar to those in MetS and T2DM [301,302]. As a result, the ability of MF to improve energy metabolism, prevent lipotoxicity and restore the redox balance in the ovaries is another mechanism of its beneficial effect on folliculogenesis and the ovulatory cycle in PCOS. In human granulosa cells, through AMPK-dependent mechanisms, MF suppresses the tumor necrosis factor-α (TNF-α)-induced production of pro-inflammatory cytokines, interleukin-8 and chemokine CXCL1/GROα [303]. The other AMPK activators, such as AICAR and Baicalin, also reduce TNF-α- and chemokine-mediated inflammatory responses in ovarian cells [303,304]. In PCOS ovarian cells, the mechanism of MF suppressive effect on the TNF-α-induced cascades includes a decrease in the gene expression of TNF-α [217].

The severity of metabolic disorders positively correlates with the therapeutic effect of MF, which is due not only to its effect on the ovaries, but also to the effect on the metabolic processes in the other tissues. A prominent feature of PCOS is decreased level of high-density lipoprotein cholesterol (HDL-C), and MF is most effective in PCOS patients with blood HDL-C levels below 50 mg/dL [226]. HDL-C modulates glucose homeostasis through AMPK-dependent mechanisms, whereby in PCOS, MF-induced AMPK activation may significantly contribute to the insulin sensitivity-restoring effect of MF [226].

### 3.5. The Sensitivity of PCOS Women to Metformin Therapy

The responsiveness of PCOS women to MF may be due to the efficiency of MF transport into the target cell and the distribution and pharmacokinetics of this drug in the tissues [28] (Figure 3). Despite the ability to freely penetrate the blood-tissue barriers and redistribute in the tissues, MF transport into the cell is carried out through the specialized transporters, including OCT1, OCT2, ATM and MATE. A decrease in their functional activity due to inactivating mutations in their genes, primarily in the *OCT1* gene, has a negative effect on the therapeutic effect of MF, in some cases making PCOS patients completely resistant to MF [151,305,306]. The incidence of *OCT1*, *OCT2* and *ATM* polymorphisms in PCOS is much higher than in women without PCOS. During genotyping of PCOS patients, non-functional alleles of the *OCT1*, *OCT2*, and *ATM* genes were detected in 29.8%, while low-functional alleles in 57.9% of cases. Non-functional alleles in the *OCT1* and *OCT2* genes were associated with poor response to MF, as well as with glucose intolerance and significantly elevated levels of proinsulin C-peptide after glucose loading [306]. In the recent work of Taiwanese scientists, the polymorphisms rs683369 (allele G) and rs628031 (allele A) were identified in the *OCT1* gene of PCOS women, and they were associated with a decrease in sensitivity of patients to both the MF and insulin therapy [151]. It should be noted that of the 38 currently identified *OCT1* gene polymorphisms, the Met^408^Val (rs628031) variant exhibits the most suppressed MF therapeutic effect [307]. Quite unexpected finding was that single-nucleotide polymorphism in the *OCT2* gene did not significantly affect the affinity to MF, although there is evidence of the involvement of the OCT2 in MF transfer into the target cells, including the ovarian cells [308].

In addition to negative effects for MF transport into the cell, a number of other factors contribute the sensitivity of PCOS patients to MF therapy. Among these factors are the severity of hyperinsulinemia and HA, the blood levels of the binding proteins for insulin, IGF-1 (IGFBP-1) and androgens (SHBG), and the blood levels of AMH and HDL-C (Figure 3).

## 4. Metformin and Gestational Diabetes Mellitus

Over the past years, the focus has been on the validity of the use of MF for the treatment of GDM and the prevention of GDM-associated preeclampsia, which is of great importance for a normal pregnancy and the birth of healthy offspring [309,310,311,312,313,314,315,316,317,318]. GDM is recognized on average in 10–20% of pregnant women, and the incidence is largely depends on diagnostic criteria and genetic, environmental and ethnic factors [319,320,321]. The dietary and exercise interventions are often used to prevent and treat GDM and improve pregnancy [317,322,323], but such interventions are effective only when used concurrently, and with pharmacological intervention [322,324]. Combining diet and exercise with vitamin D and myo-inositol supplement has been shown to be effective [324,325,326,327]. The most widely used pharmacological approach for GDM correction is insulin therapy, which is currently considered the “gold standard” of care in GDM [317,328,329]. Notably, insulin treatment leads to a number of side effects, primarily hypoglycemic episodes, dangerous for the health of pregnant women and fetus. As a result, the development of alternative pharmacological approaches is now underway, among which the use of MF and other oral hypoglycemic agents is of the greatest interest [317,330,331,332].

In 2007–2013, the first series of randomized clinical trials was carried out to assess the efficacy and safety of MF therapy in GDM, including the comparison with insulin therapy [333,334,335,336,337,338]. It was concluded that MF, to a greater extent than insulin, reduces body weight gain in pregnant women, and also reduces the frequency and severity of gestational hypertension. Unlike insulin, MF did not induce hypoglycemic episodes in the mother and fetus. At the same time, MF therapy had a little or no effect on the incidence of small for gestational age (SGA) or large for gestational-age (LGA) neonates, and had a little effect on the incidence of preeclampsia.

Subsequently, there were the additional evidences of the effectiveness of MF as a drug that reduces maternal body weight and prevents gestational hypertension [314,339,340]. Based on a large clinical data [337,341,342,343,344,345,346,347,348,349], it was concluded that MF in GDM and T2DM, to a greater extent than insulin, prevents neonatal hypoglycemia, and significantly reduces the incidence of fetuses with LGA, thereby reducing the number of newborns who require intensive care after birth [321,339,350]. It is important to note that MF therapy does not increase the rate of preterm delivery, Cesarean section, and SGA neonates [321,339,340], despite the fact that earlier studies showed the risks of MF therapy for a normal pregnancy in GDM [334,351]. Polish scientists found some increase in triglyceride levels and atherogenic index in the third trimester of pregnancy in MF-treated women with GDM. At the same time, they indicated that short-term treatment with MF and insulin had a similar impact on lipid markers of MetS in women with GDM [352].

Several advantages of MF therapy in GDM treatment were demonstrated when comparing MF efficiency with that of other oral hypoglycemic agents, including glibenclamide and its analogues [330,347,353,354,355]. The use of glyburide, which, like MF, provides adequate glycemic control in GDM, is associated with an increased risk of neonatal hypoglycemia, high neonatal birth weight and macrosomia [330,353,355].

As noted above, there is evidence that MF reduces preeclampsia in GDM, and this is based on the ability of MF to prevent endothelial dysfunctions via normalizing the levels of the anti- and pro-angiogenic factors and improving the mitochondrial energy and biogenesis [313,356,357]. MF decreases the production of anti-angiogenic factors, including soluble forms of receptors of angiogenic factors. Among them, there are the placenta-produced soluble vascular endothelial growth factor receptor-1 (VEGFR-1), also called soluble fms-like tyrosine kinase-1 (sFlt-1), which specifically binds to vascular endothelial growth factor-A (VEGF-A) and placental growth factor (PlGF), and the soluble endoglin, a soluble isomer of endoglin, which inhibits the specific binding of TGF-β1 to its receptor [313,356,357,358]. Moreover, MF and its combination with other drugs (esomeprazole, sulfasalazine) lead to an increase in the expression of VEGF-A and placental growth factor (PlGF), both powerful activators of angiogenesis, and to a decrease in TNFα-induced expression of endothelin-1, a potent vasoconstrictor [356,357]. In pregnant women, the increased activity of anti-angiogenic factors induces the systemic endothelial dysfunctions and vasospasm and provokes preeclampsia [358]. MF treatment of mice with a preeclampsia-like model induced by a six-week high-fat diet (HFD) before pregnancy reduces the blood pressure, prevents the proteinuria, normalizes the fetal and maternal weight, and leads to an improvement of the placental labyrinth and fetal vascular development, impaired in the conditions of preeclampsia [359]. This is based on MF-induced normalization of placental production of MMP-2 and VEGF in preeclamptic mice, which, in turn, improves the pregnancy outcomes [359].

In GDM, MF-induced normalization of angiogenesis and blood circulation prevents hypoperfusion of the placenta. In this regard, it should be noted that hypoperfusion leads to placental hypoxia and ischemia and oxidative stress at the maternal-fetal interface, and then the inflammatory factors and the oxidized and degraded biomolecules are transferred into the mother’s bloodstream, causing preeclamptic symptoms [360,361]. Another mechanism for the prevention of preeclampsia may be a MF-induced decrease in IR and normalization of insulin levels in women with GDM, since there is an evidence that a decrease in insulin sensitivity is an important factor provoking preeclampsia [362,363].

It is also assumed that in GDM the protective effect of MF on preeclampsia may be due to its effect on the sensitivity of the placenta to human chorionic gonadotropin (hCG) during the early pregnancy. hCG is the most important regulator of placental development and angiogenesis and is able to influence the ratio of PlGF and sFlt-1 receptors in the blood and, thereby, control the activity of pro- and anti-angiogenic factors [364]. During early pregnancy, high levels of hCG are associated with the elevated sFlt-1/PlGF ratio and the increased risk of preeclampsia [364]. The pregnant women with GDM are more likely to deliver by Cesarean section. The success of Cesarean section is significantly increased with the combined use of MF and insulin and is higher than with the use of insulin or MF alone [365]. The combination MF plus insulin had an improving effect on the delivery of LGA infant as compared to insulin monotherapy. These data indicate the need for the combined use of MF and insulin in treating GDM women with high-risk pregnancy and recommendation for Cesarean section [365].

When using MF to treat GDM, it is important to take into account the ability of this drug to easily cross the placenta barrier to affect the embryo and developing fetus [366]. At the same time, the available data do not support the teratogenic effect of MF when used during pregnancy [315,345,346,351,353,367,368,369,370,371]. Moreover, a number of positive effects of prenatally administered MF on fetal development and functional indices of newborns in GDM were demonstrated [366,372,373]. In GDM, the side effects of MF, as in the treatment of PCOS and T2DM, are mainly limited to gastrointestinal dysfunctions [287,334,374,375].

The fact that, in GDM, MF therapy in about half of cases has little or no effect [334,376], may be due to the genetic and ethnically defined environmental ethnic features, as well as the severity and pattern of hormonal and metabolic disorders in women with GDM. In this regard, the situation is similar to that in PCOS women, where the factors that determine the effectiveness of MF therapy are hyperinsulinemia, HA, functional activity of the gonadotropin and insulin/IGF-1 signaling systems in the ovaries, lipids spectrum, expression and activity of inflammatory factors and antioxidant enzymes, and also organic cation transporter-dependent transfer of MF into the target cells (see Figure 3).

To fully assess the response of pregnant women with GDM to MF, the effect of this drug on glucose homeostasis should be investigated. The glucose level one hour after the start of the oral glucose tolerance test above 212 mg/dL and the mean fasting glucose level during the first week of MF therapy above 95 mg/dL are both reliable indices for supplementing MF therapy with insulin [376]. The differences in responsiveness to MF therapy are believed to explain the results for a number of recent studies and meta-analyzes that did not show a significant effect of MF on pregnancy and its outcomes in women with GDM and on the development of preeclampsia. Based on this, a number of authors have expressed the opinion that insulin is preferable to MF in GDM treatment [377,378,379].

## 5. Metformin Treatment of Women with Diabetes Mellitus and Obesity

MF, the main oral hypoglycemic agent for the treatment of T2DM, is also widely used to treat women with T2DM before pregnancy. At the onset of pregnancy, for a significant proportion of diabetic women, on average from 11.4% to 62.5%, MF therapy is replaced by insulin therapy, which is still considered more acceptable for glycemic control in pregnancy [380,381]. However, with the accumulation of clinical data, the validity increases for the use of MF to treat women with T2DM during pregnancy, mainly due to the optimization of strategy of MF use and strong evidence of its prenatal safety [341,343,346,349,370,375,381,382,383,384,385,386,387].

As in the case of PCOS and GDM, the effectiveness of MF in the treatment of pregnant women with T2DM depends on a number of factors, including individual susceptibility to this drug. Reduced response to MF requires the use of insulin supplements to improve glycemic control. The number of pregnant T2DM women receiving MF treatment, which require supplementary insulin therapy varied significantly and ranged widely from 4% [385] to 43% [341] and even up to 84% [346]. Supplementary insulin therapy usually begins on average after the 25th week of gestation [346]. The need to add insulin is due to the ineffectiveness of glycemic control using MF alone, even with a significant increase in the daily doses of this drug. Egyptian scientists have shown that in 37% and 39% of pregnant women with T2DM, MF in the daily doses of 1500 and 2000 mg, respectively, provided a good glycemic control, while in 24% of pregnant diabetic women the use of a daily dose of 2000 mg did not secure reliable glycemic control, thus these patients required supplementary insulin treatment [343].

In contrast to women with T2DM, the use of MF in obese pregnant women who do not have severe IR, hyperglycemia and dyslipidemia is not effective. Most clinical studies indicate that MF does not significantly affect the pregnancy and the health of newborns in obese pregnant women [318,388,389,390]. The use of MF in pregnant women with overweight or obesity at the end of the first and beginning of the second trimesters does not significantly improve the outcomes of pregnancy and childbirth [390]. Although some meta-analyzes show that MF treatment of obese women can reduce maternal gestational weight gain and prevent preeclampsia, without negatively affecting the fetus, including stillbirth and congenital abnormalities [314,388], there is currently no reliable reason to recommend MF for the prevention of adverse pregnancy outcomes in obese or overweight women without pronounced signs of T2DM [388,390].

Women with type 1 diabetes mellitus (T1DM) are also characterized by an increased risk of developing preeclampsia and adverse pregnancy outcomes, and these risks are significantly higher than those in T2DM [391,392,393,394,395]. In this regard, it becomes necessary to reduce these risks, for which, traditionally, insulin therapy is used, both in the period before and during pregnancy. Adequate insulin therapy for women with T1DM before pregnancy reduces the risk of hypertensive disorder of pregnancy [396].

However, a long-term therapy with increasing doses of insulin leads to IR and hypoglycemic episodes, and also increases the mother’s body weight, thereby worsening the outcome of pregnancy [397,398,399]. In the third trimester of pregnancy, the daily insulin dose can increase to 1.0 IU/kg, and the increase in this dose can occur rapidly [400]. Weight gain in pregnant women with T1DM, which is more pronounced than among non-diabetic patients, and persistent hyperglycemia lead to a higher risk of fetal overgrowth, resulting in LGA and macrosomia [401,402].

Optimization of insulin therapy using automated insulin delivery is one of the approaches to prevent IR and normalize the body weight and glucose homeostasis [399,403,404]. Another, alternative, approach is the use of MF therapy to correct hyperglycemia in pregnant women with T1DM, especially since MF has been successfully used for a long period of time to correct hyperglycemia in non-pregnant patients with T1DM, which makes it possible to increase insulin sensitivity and reduce the insulin dose [405,406,407,408,409]. The data on MF treatment of pregnant women with T1DM are encouraging, demonstrating improved glycemic control without the need to increase the insulin dose and reducing maternal weight gain and the risk of neonatal hypoglycemia [410].

## 6. Metformin and the Male Reproduction

### 6.1. Effects Metformin on the Male Reproduction in Metabolic Disorders

Men with T2DM, MetS and severe obesity have hypogonadotropic hypogonadism and other reproductive dysfunctions that are usually associated with impaired spermatogenesis and steroidogenesis [411,412,413,414,415,416,417,418,419,420,421,422]. The main factors that negatively affect the spermatogenesis and steroidogenesis in these metabolic disorders are dyslipidemia, IR, hormonal dysregulation in the Leydig and Sertoli cells, imbalance of cytokines in the testes, and an increase in the inflammation and the apoptotic and oxidative processes in testicular and germ cells [414,418,423,424,425].

A recent meta-analysis of 11 clinical studies involving 1731 men with MetS showed a significant decrease in total sperm count, sperm concentration, sperm survival, and the number of sperm with normal morphology and progressive motility, as well as an increase in sperm DNA fragmentation and functional changes in the mitochondrial energy and biogenesis [426]. Along with an androgen deficiency, the blood levels of FSH and inhibin B were significantly decreased in men with MetS, while the blood levels of LH, estrogen, and AMH were changed to a small extent [426].

In contrast to patients with MetS, in T2DM, the results of studies of the sperm parameters, androgenic status and gonadotropins levels differed greatly, which was due to the age heterogeneity, differences in the severity and duration of T2DM, and the features of the etiology and pathogenesis of this disease [417]. Some authors showed the pathological changes in spermatogenesis, including a decrease in the number, motility and survival of spermatozoa and an increase in their defective forms and DNA fragmentation [418,427]. Meanwhile, a meta-analysis carried out by Greek scientists in 2016 indicates a decrease in seminal volume and the number of motile spermatozoa in men with T2DM, while the total number of sperm and their normal forms did not change significantly [417]. It can be assumed that the arterial hypertension, typical for MetS and T2DM, negatively affects morphology of the seminiferous tubules and spermatogenesis. Using the rat models, it was shown that arterial hypertension, interfering with normal microcirculation of blood in the testes, led to the impaired sperm maturation [428].

Since MF improves the lipid and carbohydrate metabolism, increases insulin sensitivity, prevents overproduction of reactive oxygen species and pro-inflammatory factors in men with MetS and T2DM, the use of MF therapy should thereby have a restorative effect on reproductive functions in men with these metabolic disorders [94,429]. Indeed, there are the experimental and clinical evidences of a positive effect of MF on the male reproductive dysfunctions in MetS and T2DM, but the mechanisms and the factors influencing the effectiveness of the restorative MF action are still poorly understood [94,430,431]. In addition, a number of authors did not confirm the improving effect of MF on male reproduction or even showed its negative effect on androgenic status in men with severe obesity and T2DM [432,433,434].

In addition to the MF effects in the testes, which is well documented (see the Section 6.2 and Section 6.3 for details), the targets of MF can also be hypothalamic GnRH-expressing neurons and pituitary gonadotrophs, the other links of the male HPG axis. However, there are currently no data on the direct influence of MF on GnRH-expressing neurons and gonadotrophs in men. At the same time, it is known that MF, when acting on the primary cultures of pituitary cells, is able to influence the expression of LH and FSH, thereby controlling the synthesis of gonadotropins in them [435].

### 6.2. The Clinical Studies of the Metformin Efficacy to Treat Reproductive Dysfunctions in Men

The first reliable evidence of the efficacy of MF therapy in improvement of the reproductive functions in men with MetS was presented in 2010 [436]. Four-month therapy of 35 men with MF (1700 mg/day), which was combined with a balanced normocaloric diet and physical activity, led to a decrease in insulin levels and the HOMA-IR index and an increase in the blood levels of total and free testosterone. Additively, in men with hypogonadotropic hypogonadism, the MF-induced normalization of FSH levels was shown [436]. These data provided evidence of a restorative effect of MF on the hormonal status of the HPG axis in men with MetS and androgen deficiency.

Subsequently, Giuseppe Morgante and colleagues studied the restorative effect of MF (850 mg/day for the first week, 1700 mg/day for the second week, and 2550 mg/day for the rest six-month period) on spermatogenic function and hormonal indices of the HPG axis in 45 men with MetS who had the impaired spermatogenesis, including oligospermia, teratozoospermia, and asthenozoospermia. MF increased the number of spermatozoa, improved their motility and morphology and, thereby, partially restored fertility [430]. In MF-treated men, the blood testosterone and LH levels were significantly increased, while the levels of estradiol and SHBG were decreased, which led to an increase in the testosterone/estradiol ratio and an improvement of androgenic status. The normalization of LH levels induced by MF indicates the restoration of the hypothalamic mechanisms responsible for pulsatile LH release [430]. Since an improvement in spermatogenesis and the hormonal status of the HPG axis was associated with a 43% decrease in the HOMA-IR index, this confirms the leading role of IR and glucose intolerance in development of spermatogenesis dysfunctions and androgen deficiency in men with MetS.

One clinical study investigated the effect of a combined three-month treatment of men with impaired glucose tolerance or with T2DM using MF at a daily dose of 2000 mg and CC at a low dose (25 mg) on their androgenic status [434]. The CC is characterized by the ability to significantly improve hormonal parameters in men with hypogonadotropic hypogonadism, primarily by normalizing the secretion of gonadotropins [437,438]. The combined therapy with MF plus CC led to an increase in the blood testosterone, LH and FSH levels, while MF monotherapy in this case was ineffective [434]. Quite unexpectedly, the authors did not find a significant effect of MF monotherapy on metabolic parameters, despite the improvement in patients’ insulin sensitivity [434].

Since MF reduces body weight in obese patients, and obesity is associated with androgen deficiency and impaired spermatogenesis [411,415,422,439,440,441,442,443], there is every reason to believe that one of the mechanisms of the improving effect of MF on male reproduction may be the normalization of adipokine status. This is supported by evidence of a positive effect of bariatric surgery on the blood levels of testosterone, LH and FSH in obese and MetS men via reduction in the adipose tissue mass and normalization in the energy metabolism [444]. Normalization of body weight in the late adolescence is important for normal puberty, since overweight and obesity during this period lead to impaired testicular function in the reproductive age [445]. Therefore, it is worthwhile to evaluate the possibility of using MF therapy for prevention of reproductive disorders in adolescents with obesity and MetS.

At the same time, there are studies showing a decrease in testosterone levels in men with T2DM and obesity during MF therapy [432,433]. Turkish scientists found that treatment of such patients with MF (1700 mg/day, three months) led to an increase in SHBG levels and a decrease in free testosterone levels in the blood [432]. However, in this study, MF-treated patients with T2DM and obesity received a low-calorie diet, which could have a negative influence on the functional state of the HPG axis and lead to a decrease in testosterone production. Conflicting clinical results on the effect of MF therapy on male reproduction require further research, especially considering the encouraging results obtained in animals with experimental models of metabolic disorders, as well as the proven ability of some other antidiabetic agents to restore androgenic status in diabetic pathology and MetS [446,447].

### 6.3. The Experimental Studies of Metformin Effects on Male Reproductive Dysfunctions in Animal Models of Metabolic Diseases

Most experimental works point to the restorative effect of MF on spermatogenesis and testicular steroidogenesis in animals with the experimental models of MetS, DM and obesity [448,449,450,451,452,453,454,455]. Treatment of male rats with streptozotocin (STZ)-induced diabetes with MF (100 and 500 mg/kg) for 4 or 8 weeks restored the antioxidant system and redox balance in the testes, normalized the proliferative activity of testicular somatic and germinal cells, prevented the DM-induced instability of their genome, thereby exerting an antigenotoxic effect [448]. The effect of MF was dose-dependent, and this drug, even at a relatively high dose, did not cause genotoxic and cytotoxic effects on the testicular cells of both diabetic and control animals [448].

The treatment of male Wistar rats with severe T2DM with pioglitazone (1 mg/kg) and low-dose MF (50 mg/kg) for four weeks resulted in a decrease in defective sperm and increased the caudal sperm count [449]. The treatment of male Sprague-Dawley rats with STZ-induced DM with pioglitazone alone did not prevent destructive changes in the testes, while MF was effective in this regard [456], which indicates a key role of MF in the restoration of spermatogenesis in combined therapy with pioglitazone and MF.

The key role in the development of testicular dysfunctions and in the deterioration of spermatogenesis and steroidogenesis belongs to the activation of oxidative stress, inflammation and apoptosis in the testes. Since MF has pronounced anti-inflammatory, antioxidant and antiapoptotic effects on testicular cells, these effects may be largely due to the MF-induced restoration of spermatogenesis and steroidogenesis in experimental MetS and DM [431,450,452,453,454,457]. The treatment of male Sprague-Dawley rats with MetS using MF (100 mg/kg/day, 8 weeks) weakens the apoptotic and pro-inflammatory processes in the testes and, thereby, increases the number of spermatogonia, Sertoli and Leydig cells and motile spermatozoa, decreases the number of small, atrophic and distorted seminiferous tubules, and improves the morphology of the seminiferous tubules [450]. MF also increases the blood testosterone levels, which are significantly reduced in diabetes, and also normalizes the blood levels of insulin, leptin, and estrogens [450].

A pronounced anti-inflammatory, antioxidant and antiapoptotic effect of MF was demonstrated in the testes of Sprague-Dawley rats with STZ-induced DM, which were treated with MF at a daily dose of 300 mg/kg/day for 4 weeks [452,457]. The anti-inflammatory effect of MF was due to a reduction in the expression of the inflammatory factors NF-κB, TNFα and interleukin-1β, increased in DM, as well as the restoration of the expression interleukin-10 with anti-inflammatory activity [452]. The antioxidant effect was based on a partial restoration of the expression of the main antioxidant enzymes, such as superoxide dismutase, catalase, and glutathione peroxidase, a decrease in the level of malonic aldehyde, the marker of oxidative stress, as well as a normalization of the expression of Nrf2 factor, which controls the expression of the enzymes protecting the cell from reactive oxygen species. The antiapoptotic effect of MF included a decrease in the DM-induced expression of proapoptotic protein p53 and the activity of caspases-3 and -8, as well as the restoration of the expression of antiapoptotic protein Bcl-2 and its ratio with proapoptotic protein Bax, reduced in diabetes [452]. In MF-treated diabetic rats, an increase in the blood and testicular levels of testosterone, a normalization of the number of the Leydig cells, an improvement in sperm morphology, a decrease in sperm nuclear DNA fragmentation and restoration of the expression and activity of the transport protein StAR and steroidogenic enzymes, such as the cytochrome P450scc (CYP11A1) and the dehydrogenases 3β-HSD and 17β-HSD were shown [452,457]. MF also increased the testicular levels of the androgenic and LH receptors, reduced in diabetes, indicating restored sensitivity of testicular cells to androgens and gonadotropins [452].

The antioxidant effect of MF was critical for its restorative effect on spermatogenesis and steroidogenesis in C57BL/6 mice with MetS induced by a HFD and cholesterol-rich diet [431]. A high content of cholesterol in food led to an increase in intratesticular cholesterol concentration, provoked the deposition of lipids in the seminiferous tubules, and impaired the morphology of the Leydig cells, stimulating the ER stress and apoptosis, culminating in reduction of testosterone synthesis [458,459]. Eight-week administration of MF led to a decrease in atherogenic cholesterol levels in the blood and lowered the cholesterol level in the testes, restoring intratesticular testosterone levels, which were decreased in MetS. Along with this, in MF-treated mice, the expression of 17β-HSD was restored [431]. The 17β-HSD catalyzes the final stage of testicular steroidogenesis, the conversion of androstenedione to testosterone. Androstenedione is a precursor of not only testosterone, but also estradiol, and its accumulation in the testes in obesity and MetS, due to the weakening of 17β-HSD activity, can enhance the synthesis of estrogens [460]. Using the cell cultures, it was shown that MF is able to suppress both the basal and insulin-stimulated expression of aromatase, which converts androgens to estrogens, and thereby restore the testosterone/estradiol ratio and improve spermatogenesis [244,246]. Thus, MF triggers several mechanisms, which lead to the normalization of the balance of sex steroid hormones in the testes in metabolic disorders.

MF restores the functions of testicular cells and spermatogenesis in rats with testicular ischemia/reperfusion caused by both clipping of the left testicular artery and vein [461,462] and the testicular torsion and deformity [451]. In the case of clipping of the testicular artery and vein, MF (100 mg/kg) and its combination with melatonin restored the activity of superoxide dismutase in the testes, which was reduced in ischemia/reperfusion, and normalized the activity of myeloperoxidase and the malonic aldehyde levels [461,462]. In the case of testicular torsion, in the testes, MF reduced the malonic aldehyde levels and inhibited the activity of caspase-3, a key enzyme of apoptosis. The antioxidant effect of MF in testicular cells was detected as early as 4 h after testicular injury [451].

Along with MF monotherapy, the combinations of MF with different natural antioxidants, including honey [463] and Malaysian propolis [453,464], restored the testicular function and hormonal parameters of the male HPG axis in diabetic animals. Some compounds with antioxidant and antiapoptotic activity, but different from MF molecular mechanisms, for example, l-carnitine, may be effective [465]. There is a question in this regard, for which mechanisms determine the antioxidant and antiapoptotic effects of MF and its restorative effect on spermatogenesis and testicular steroidogenesis? There is reason to believe that AMPK-dependent pathways are the most important, since AMPK is undoubtedly the main target of MF in the testes. It is important to note that in DM, MetS and testicular ischemia-reperfusion, one of the triggers of impaired spermatogenesis and steroidogenesis is a decrease in the activity of testicular AMPK. In 2012, Ana Hurtado de Llera and colleagues showed that pharmacological inhibition of AMPK in the testes dramatically reduces the percentages of motile and rapid spermatozoa [466]. There is a lot of evidence that AMPK regulates the growth and differentiation of the Sertoli and Leydig cells, controls the motor activity of spermatozoa and their acrosomal reaction. Moreover, AMPK is responsible for antioxidant activity and production of reactive oxygen species in the testicular somatic and germ cells, and determines the metabolic processes in them, including lipid metabolism [194,467,468,469,470]. Accordingly, the normalization of AMPK-dependent pathways in the testes under the influence of MF may be the main mechanisms of its action on improving male reproduction in DM and other metabolic diseases.

It can be assumed that the normalization of AMPK signaling in the testes may be due not only to the direct effect of MF on testicular AMPK activity, but also due to the restoration of the leptin signaling pathway in the testes of diabetic rats, especially since AMPK is also one of the leptin targets [471,472], in particular in the testes [473]. The MF treatment (4 weeks, 120 mg/kg) of male Wistar rats with HFD/STZ-induced T2DM and severe hyperleptinemina led to a normalization of the blood and testicular levels of leptin and an increase in the number of testicular leptin receptors [455]. The two-week MF treatment (500 mg/kg) of albino mice with STZ-induced T1DM also improved the leptin signaling pathway in the testes, increasing the expression of ObRb, functionally active isoform of leptin receptor, in the Leydig cells, primary spermatocytes and round spermatids [454]. The improvement of the testicular leptin signaling pathway was accompanied by the restoration of the steroidogenic genes expression, including the cholesterol-transporting protein StAR, an increase in the sensitivity of the Leydig cells to hCG, and a weakening of the apoptotic processes in testicular cells [454,455].

## 7. Conclusions

The data presented in the review convincingly prove that MF has an improving effect on reproductive functions both in women with PCOS, GDM and T2DM, and in men with DM and MetS. At the same time, the effectiveness of MF therapy is due to a large number of different factors that must be taken into account when choosing this therapy and also when developing a strategy for using MF. Firstly, it is necessary to assess the efficiency of MF transport into the cell, which depends on the functional activity of the organic cation transporters and can be disrupted by inactivating mutations in their genes. The presence of certain mutations leads to a loss of responsiveness to MF and makes the use of MF therapy meaningless. Secondly, as demonstrated in women with PCOS, GDM and T2DM, the MF therapy is more effective in the severely overweight and obese patients with IR, compensatory hyperinsulinemia, impaired glucose tolerance, as well as with dyslipidemia, which is due to a decrease in the blood levels of HDL-C. This is not surprising, since the clinical effect of MF therapy is due to an improvement in insulin sensitivity, a decrease in the adipose tissue mass, and restoration of the glucose and lipid metabolism.

In addition, as demonstrated in a number of clinical studies in PCOS women, the effectiveness of MF is largely determined by the hormonal status of the ovaries and the functioning of the HPG axis. The MF therapy may be most effective in PCOS women who have: (1) severe HA, which may be due to hyperactivation of ovarian insulin/IGF-1-regulated signaling pathways that stimulate androgen synthesis, as well as an increase in LH levels and the LH/FSH ratio and a decrease in the blood levels of IGFBP-1 and SHBG; (2) an increase in the AMH production; and (3) an increased aromatase expression and FSH-induced estrogen synthesis in the ovaries. There is reason to believe that various combinations of these factors, including those with IR and metabolic disorders, may become reliable indications for prescribing MF alone and in combinations with other drugs, diet or exercises to correct the reproductive functions in PCOS. This can be helpful when using MF to treat the pregnant women with GDM and T2DM. Moreover, a systemic approach based on the analysis of the combination and severity of metabolic and hormonal dysfunctions may be useful to assess the efficacy of MF therapy for improving spermatogenesis and steroidogenesis in men with DM and MetS.

Since some of MF targets overlap well with those of leptin, the assessment of leptin status in patients with reproductive disorders may also be important. As a result, leptin resistance, both systemic and in the ovaries/testes, as well as the changes in the hypothalamic leptin signaling pathways, negatively affecting the production of GnRH, can become factors that will determine the effectiveness of MF therapy. In this regard, it should be noted that the central mechanisms of action of MF, which easily penetrates the CNS and improves the metabolism of the neuronal and glial cells, still remain underestimated. By acting on the CNS, MF restores the signaling networks of the hypothalamus and the other brain regions that are involved in the control of reproductive functions and undergo significant compensatory and pathological changes in metabolic and endocrine disorders, including PCOS, GDM, T2DM, and MetS.

A unique feature of MF is the multiplicity of molecular mechanisms of its action on target cells, which include direct or indirect regulation of the AMPK-, calcium- and cAMP-dependent signaling pathways, as well as the MAPK cascade and the IRS/PI 3-K/Akt pathway. As a result, MF controls not only energy and metabolic processes in the cell, but also the processes of growth, differentiation, apoptosis, inflammation, and ER stress. At the same time, most of the regulatory effects of MF are based largely on its modulating and normalizing influence on intracellular signaling cascades than on their prolonged stimulation or suppression. Depending on the functional state of the IRS/PI 3-K/Akt pathway, MF can either prevent its hyperactivation, which is especially important for its antitumor effect, or, on the contrary, restore its reduced activity, improving the survival of target cells and their sensitivity to insulin and leptin. As expected, MF therapy affects the responsiveness of hypothalamic neurons, pituitary gonadotrophs, and testicular and ovarian cells to the hormones, growth factors, adipokines and cytokines, but more studies are required for complete elucidation of all regulatory mechanisms involved.

The use of MF in combination with the other drugs has great potential. This is supported by the encouraging results of clinical trials of the combined therapy with MF and insulin in pregnant women with GDM and T2DM. In metabolic and endocrine disorders, the combined therapy not only allows to increase the efficiency and pattern of the effects of MF on the HPG axis, but also to reduce the pharmacological doses of drugs, including MF, thus avoiding possible side effects of high-dose drug administration, including the undesirable effect of MF on the functioning of the gastrointestinal tract.

The presented results indicate a significant and not yet fully understood potential of MF therapy for the correction of reproductive dysfunctions in women and men. Significantly, of great importance are the absence of a teratogenic effect of MF and the low risks of MF therapy on the health of the mother and child. It should be taken into account that the unjustified use of MF for the treatment of patients lacking profoundly manifesting metabolic and endocrine disorders can lead to energy and metabolic imbalance and further deterioration of the functional state of their reproductive system.

## Figures and Tables

**Figure 1 pharmaceuticals-14-00042-f001:**
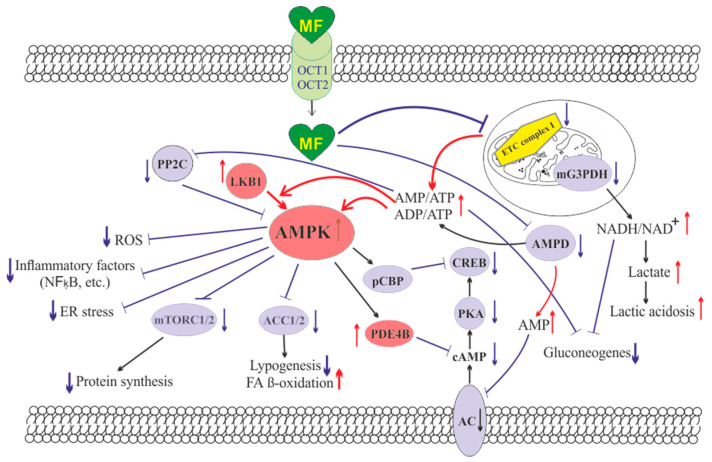
The cellular mechanisms of metformin action which are carried out by activation of the AMP-activated protein kinase and inhibition of the mitochondrial electron transport chain complex I. Abbreviations: AC, adenylyl cyclase; ACC1/2, acetyl-CoA carboxylases 1 and 2; AMPD, AMP deaminase; AMPK, the heterotrimeric AMP-activated protein kinase consisting of the α1/2 (the target for activation phosphorylation at the Thr^172^), β1/2 and γ1/2/3 subunits; CREB, cAMP-activated transcription factor (cAMP response element-binding protein); ETC complex I, the mitochondrial NADH-dehydrogenase complex, the first complex of the respiratory electron transport chain; FA, fatty acids; LKB1, liver kinase B1; mG3PDH, mitochondrial glycerol-3-phosphate dehydrogenase; mTORC2, the mTOR complex 2; NFκB, nuclear factor κB; OCT1/2, the organic cations transporters 1 and 2; pCBP, the Ser^436^-phosphorylated form of CREB-binding protein with acetyltransferase activity, a co-activator of the factor CREB; PDE4B, cAMP-specific 3′,5′-cyclic phosphodiesterase 4B; PKA, cAMP-dependent protein kinase; PP2C, protein phosphatase 2C; ROS, reactive oxygen species.

**Figure 2 pharmaceuticals-14-00042-f002:**
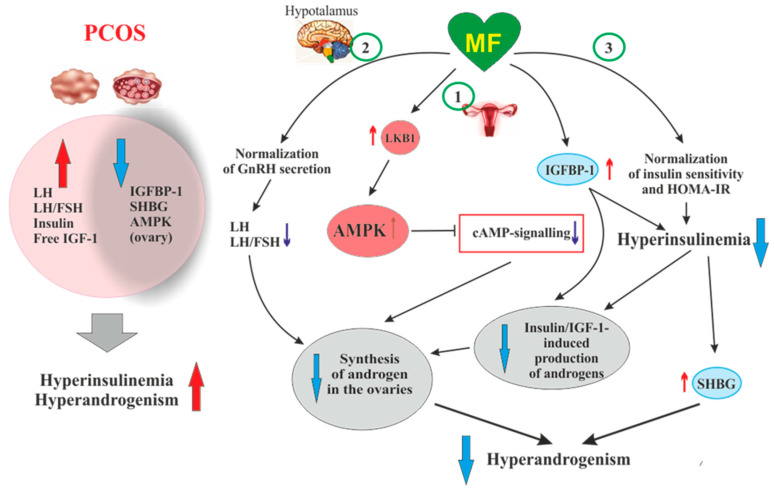
The pathways involved in the inhibitory effect of metformin on hyperandrogenism in PCOS. Hyperinsulinemia and HA are among the key pathogenetic factors in the development of PCOS, which is why, their attenuation by MF is the most important mechanism for improving effect of this drug on ovarian function in PCOS women. In PCOS, MF-induced increase in insulin sensitivity leads to a decrease in the HOMA-IR and a weakening of compensatory hyperinsulinemia. Another mechanism for lowering insulin levels may be an increase in the level of IGFBP-1, which specifically binds insulin and IGF-1. In PCOS, the expression of IGFBP-1 is generally reduced, and MF treatment may be one way to normalize it. A reduced hyperinsulinemia and an increase in IGFB-1 levels lead to a decrease in the stimulating effect of insulin and IGF-1 on the ovarian steroidogenesis and a weakening of HA. Hyperinsulinemia leads to a decrease in the production of SHBG, which provokes HA in PCOS. MF-induced reduction of hyperinsulinemia leads to the normalization of the SHBG levels, thereby preventing excess androgen levels in the blood. By improving the functionality of the hypothalamic signaling network responsible for the pulsatile secretion of GnRH, treatment with MF leads to the normalization of blood LH levels and the LH/FSH ratio, both of which are increased in PCOS. A decrease in blood LH levels results in a weakening of gonadotropin-induced androgen production by the ovaries. A direct regulatory effect of MF on ovarian steroidogenesis was also established. By inhibiting the mitochondrial ETC complex I, stimulating the LKB1 activity and, as a result, increasing the AMPK activity, MF reduces the synthesis of androstenedione in the ovarian cells and prevents HA. It can be assumed that the prevalence of some mechanisms of the inhibitory effect of MF on HA is due to the characteristic features of PCOS pathogenesis and the metabolic and hormonal status of the ovaries. Details and bibliographic references are presented in the Section 3.4. Abbreviations: AMPK, AMP-activated protein kinase; FSH, follicle-stimulating hormone; HA, hyperandrogenism; HOMA-IR, homeostasis model assessment of insulin resistance; IGF-1, insulin-like growth factor-1; IGFBP-1, insulin-like growth factor-binding protein-1; LH, luteinizing hormone; LKB1, liver kinase B1; SHBG, androgen and sex hormone-binding globulin.

**Figure 3 pharmaceuticals-14-00042-f003:**
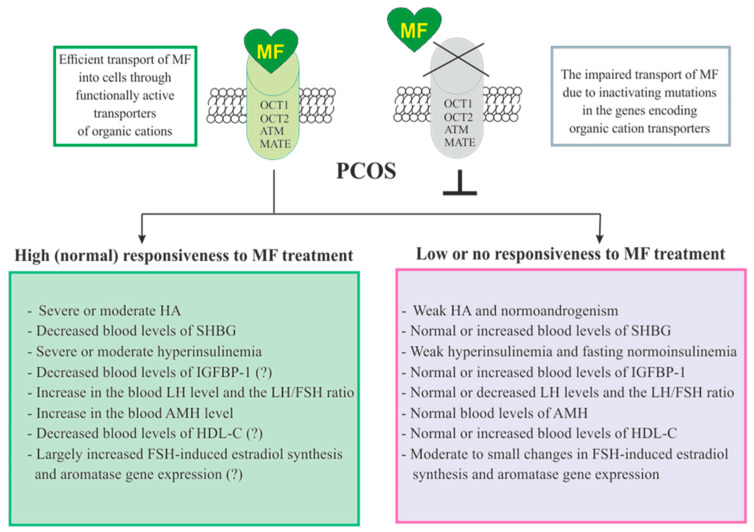
Factors determining responsiveness to metformin and the effectiveness of metformin therapy in women with PCOS. *W*omen with PCOS, as well as the patients with other pathologies, must have functionally active transporters of organic cations (OCT1, OCT2, and others) in order to respond to MF, since inactivating mutations and polymorphisms in the genes encoding these transporters lead to impairment of MF transport into the cell and make MF therapy ineffective. Since MF improves metabolic parameters and insulin sensitivity, its effectiveness in PCOS women with overweight or obesity, as well as with severe dyslipidemia and impaired glucose tolerance, is usually higher. There is evidence that MF therapy is most effective in PCOS women who have pronounced signs of hyperinsulinemia and hyperandrogenism, the increased LH levels and the LH/FSH ratio, the decreased levels of SHBG, IGFBP-1 and HDL-C, and the increased levels of AMH. It can also be assumed that MF will be more effective in patients with increased aromatase expression and ovarian hypersensitivity to FSH, since one of the mechanisms of MF action is normalization of the expression of genes encoding the FSH receptor and aromatase, as well as normalization in the response of ovarian cells to stimulation of FSH. Details and bibliographic references are presented in the Section 3.5. Abbreviations: AMH, anti-Müllerian hormone; FSH, follicle-stimulating hormone; HDL-C, high-density lipoprotein cholesterol; IGFBP-1, insulin-like growth factor-binding protein-1; LH, luteinizing hormone; OCT1 and OCT2, organic cation transporters-1 and 2; SHBG, androgen and sex hormone-binding globulin.

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
