# Peer review of "Improvement Effect of Metformin on Female and Male Reproduction in Endocrine Pathologies and Its Mechanisms"

_pharmaceuticals, 2021, doi:10.3390/ph14010042_

Round 1
Reviewer 1 Report
The review by Shpakov AO is quite informative and mainly centered in female reproduction and PCOS, but also with some data about men reproduction. Also discuss som data of the gestational use of metformin. The review is valuable and has interesting information about molecular effects of metformin. However, in some parts it is redundant and tedious. Some explanations should be more concrete. A weakness is that in several parts it is not clear if the effects are attributable to Metformin acting on its molecular targets or they are a consequence of improving metabolism and/or follicular development. An English language revision is encouraged to improve word choice and fluidity.
1) Please send the manuscript to an English native speaker to improve the fluidity of the manuscript.
2) Information in L62-L66 is remarkably similar to information in L87-88. Please try not being redundant.
3) In L62-74 the author states that the main target of MF is AMPK. But in the paragraph remains unclear whether MF binds AMPK or not. Please be clear, if MF binds AMPK or only modifies AMP/ATP ratio trough the action on electron transport chain at mithocondrion.
4) In L 87-114 the author provides a lot of information, but the section lacks bibliographical references. Please insert references. Here the author refers to Complex I of ETC as the main target of metformin, while above refers to AMPK as the main target.
5) It is not clear whether metformin directly blocks PPC2, binds and directly activates AMPK, binds and activate LKB1 or directly activates PDE4B.
6) In line with the previous point, please indicate somehow in Figure 1 when MF binds directly a target or when activates/inhibits indirectly a target or process.
7) L130-134 It is not clear again whether the mentioned low dose effect of MF (Apparently independent of Complex I ETC action) requires its binding to AMPK.
8) Please define “CBP” in the text.
9) The entire paragraph of mechanism of MF has various redundant statements. Please reorder to not repeat information.
10) L198. It is not common refer to PCOS as a polyendocrine syndrome.
11) L220-222. Please add a reference supporting the phrase.
12) In 3.2 it is not clear what spectrum of treatment of PCOS will be discussed. What PCOS treatment means?
13) L302-316 refer to treatment of hyperglycemia more than specifically PCOS.
14) L336-337 please rewrite the sentence, it is complex to understand.
15) L349 Please correct Folluicle-stimulating Hormone
16) L364-383 lack of references, and some sentences require bibliographical support (See the next point)
17) L 364 Why it is stated that IR and androgen excess are the main pathogenic factors of PCOS? This is a complex affirmation due the uncertain PCOS pathogenesis. I agree that most of PCOS patients have IR and HA, but it is not clear that both factors are the cause of PCOS.
18) L239-245 and L559-566 show the same information. Please not being redundant.
19) It is a little bit confusing that author say that MF increases the sensitivity to Gonadotropins but decreases the effects of CREB mediated transcription, since both LH and FSH has a Gs/AMPc/PKA/CREB mechanism.
20) Regarding AMH (3.4.5) please clarify if MF effect on AMH is direct (For example through AMPK action) or depends on normalization of follicular development by other mechanism. With a little sentence at the end of the section is enough.
21) L685-685. Please be cautious with the affirmation that something …is the main pathogenic mechanism... It is better to say for example “oxidative stress and inflammation are essential/crucial/very important/fundamental in the pathogenesis of PCOS”
22) L732-744 lack o references and repeat some previous ideas. If it is a conclusion paragraph, I suggest only a simple 2-3 lines sentence with the conclusion of the section.
23) L783-796 What is the mechanism by which metformin regulate vascular growth factors?
Author Response
RESPONSE TO REVIEWER 1
Alexander O. Shpakov «Improvement effect of metformin on the female and male reproduction in endocrine pathologies and its mechanisms» (1016034)
COMMON SECTION OF RESPONSE TO REVIEWERS
I am very grateful to the Reviewers for a detailed analysis of the review article and for the comments made. I sincerely hope that the explanations and changes that I have made based on the comments and remarks of the Reviewers have improved the review article.
In accordance with the requirements of the Reviewers, I significantly revised the review article and made the following main changes and additions to it (in more details, the changes and corrections are presented below in the extended answers to the questions and comments of Reviewers 1 and 2).
- According to the comments of both reviewers, the Section 2 “Brief description of the targets and molecular mechanisms of action of metformin” has been extensively revised, restructured and supplemented with the necessary bibliographic references and explanations. In the process of revision of the Section 2, duplicate or redundant information was removed (see lines 51-155 in the manuscript).
- Unfortunately, due to inadequate text formatting (technical error), the descriptions (figure captions) for the Figures 2 and 3 began to be considered as parts of the main text. In this regard, these “redundant boxes” broke the logical presentation of the data and their discussion in the Sections 3.4 and 3.5, and began to be perceived as the conclusions to the corresponding sections. There were no bibliographic references in these “redundant boxes”, which led to the Comments 16 and 22 by Reviewer 1 and a general comment by Reviewer 2. These descriptions (figure captions) are now properly formatted and do not mix with the main text of the review.
- In accordance with the comments of the reviewers, the Section 6 "The effect of metformin on the offspring" has been removed from the review along with two tables. This is due to the fact that this section is not directly related to the general topic of the review, although it is of significant interest for medicine. I hope that removing the Section 6 improves the structure of the overview article and makes it easier to read. All sections have been modified as necessary due to deletion of the Section 6.
According to requirement of the Editor and Reviewers, all changes and corrections to the text are highlighted in yellow.
In conclusion, I would like to thank the Reviewers once again for the comments that helped improve the review article.
With best regards,
The Author
REVIEWER 1
Common comments
The review by Shpakov AO is quite informative and mainly centered in female reproduction and PCOS, but also with some data about men reproduction. Also discuss some data of the gestational use of metformin. The review is valuable and has interesting information about molecular effects of metformin. However, in some parts it is redundant and tedious. Some explanations should be more concrete. A weakness is that in several parts it is not clear if the effects are attributable to Metformin acting on its molecular targets or they are a consequence of improving metabolism and/or follicular development. An English language revision is encouraged to improve word choice and fluidity.
Response/corrections
I agree with the reviewer and when finalizing the review, all comments on its structural organization were taken into account (more details above: (i) complete revision of section 2, (ii) deletion of section 6, and (iii) correction of formatting of figure captions to avoid their inclusion in the main text).
1) Please send the manuscript to an English native speaker to improve the fluidity of the manuscript.
Response/corrections
I coordinated the English language with a native English speaker and made the necessary changes (highlighted in yellow).
2) Information in L62-L66 is remarkably similar to information in L87-88. Please try not being redundant.
Response/corrections
In accordance with the reviewer's comment, the sentence with repetitive information on page 3 (lines 87-88) has been removed, and the sentence on page 2 (lines 54-57*) has been modified:
“The molecule of MF, a small hydrophilic cation, it is transported from the extracellular space to the cytoplasm of target cell through organic cation transporters-1 and -2 (OCT1, OCT2), multidrug and toxin extrusion transporters (MATE), and ATM (ataxia telangiectasia mutated) transporter, and OCT1 and OCT2 are considered main functional units of MF transmembrane transport [28]”
* - Hereinafter, the Response / corrections column presents the new line and page numbering, while the Comments column retains the old numbering. The text from the new version of the article is given in quotation marks.
3) In L62-74 the author states that the main target of MF is AMPK. But in the paragraph remains unclear whether MF binds AMPK or not. Please be clear, if MF binds AMPK or only modifies AMP/ATP ratio trough the action on electron transport chain at mitochondrion.
Response/corrections
Metformin indirectly activates AMPK through an increase in both the AMP levels and the AMP/ATP ratio, which is indicated in the text (see below, lines 59-61).
The paragraph on the effect of metformin on mitochondrial complex 1 of the electron transport chain has been rewritten.
“The ultimate intracellular target for MF is the 5′-adenosine monophosphate-activated protein kinase (AMPK), the key energy sensor of the cell, although MF does not interact directly with the enzyme [22, 30-32]”
4) In L 87-114 the author provides a lot of information, but the section lacks bibliographical references. Please insert references. Here the author refers to Complex I of ETC as the main target of metformin, while above refers to AMPK as the main target.
Response/corrections
The paragraph corresponding to lines 87-114 (old version), like the Section 2 “Summary of cell targets and molecular mechanisms of action of metformin”, has undergone a total revision, and the necessary bibliographic references have been added (see lines 51-155 in the manuscript).
5) It is not clear whether metformin directly blocks PPC2, binds and directly activates AMPK, binds and activate LKB1 or directly activates PDE4B.
Response/corrections
Metformin increases the level of AMP, which suppresses the activity of phosphatase PPC2 and, thereby, increases the activity of AMPK (the corresponding explanation has been made in the text, lines 78-82).
“A negative regulator of AMPK is the protein phosphatase 2C (PP2C), which dephosphorylates and inactivates the α-subunit of AMPK, causing the dissociation of the αβγ-heterotrimeric complex. Elevated levels of AMP lead to an inhibition of PP2C activity, which allows AMPK to remain stable in the active Thr172-phosphorylated state [49, 50]”.
Metformin activates AMPK, which carries out phosphorylation and activation of phosphodiesterase 4B, thereby reducing the level of cAMP in the cell and weakening the hormone-induced cAMP signaling cascades (the corresponding explanation has been made in the text, lines 120-122).
“The MF-induced activation of AMPK promotes phosphorylation and activation of cAMP-specific 3’,5’-cyclic phosphodiesterase 4B (PDE4B), thereby reducing the intracellular level of cAMP [68]”
6) In line with the previous point, please indicate somehow in Figure 1 when MF binds directly a target or when activates/inhibits indirectly a target or process.
Response/corrections
Both in the text and in the Figure 1, it is now explained/shown which targets the metformin activates directly (ETC1), and which indirectly: either through an increase in the AMP level (LKB1 kinase), or through the AMPK activation (cAMP-specific phosphodiesterase PDE4). In the Figure 1, it has also been shown that inhibition of phosphatase PPC2 occurs indirectly through MF-induced increase in intracellular AMP level.
7) L130-134 It is not clear again whether the mentioned low dose effect of MF (Apparently independent of Complex I ETC action) requires its binding to AMPK.
Response/corrections
The Section 2 “Summary of cell targets and molecular mechanisms of action of metformin” has undergone a total revision, and the necessary explanations are included in the text (see lines 51-155 in the manuscript).
8) Please define “CBP” in the text.
Response/corrections
Thank you very much for your comment, since the transcript for CBP is given only in the caption to the Figure 1, and it is not in the text. Accordingly, a decoding for this factor was introduced into the text (line 127-129):
“Furthermore, MF-induced AMPK activation induces the protein kinase ι/λ-mediated phosphorylation of cyclic AMP response element binding (CREB)-binding protein (CBP or CREBBP)…”
9) The entire paragraph of mechanism of MF has various redundant statements. Please reorder to not repeat information.
Response/corrections
The Section 2 “Summary of cell targets and molecular mechanisms of action of metformin” has undergone a total revision, and the necessary explanations are included in the text (see lines 55-155 in the manuscript).
10) L191. It is not common refer to PCOS as a polyendocrine syndrome.
Response/corrections
The complexity of endocrine disorders in PCOS, including the insulin resistance, the changes in the hypothalamic and pituitary links of the gonadal axis and the levels and ratios of sex steroid hormones, as well as the changes in the functional activity of the adrenal glands, allows it to be classified as a polyendocrine syndrome. Thus, PCOS is a disease that includes the pathological changes in various components of the endocrine system.
But I completely agree with the Reviewer that this is not a generally accepted opinion, and therefore changed the corresponding phrase (lines 158-159, see below):
“The PCOS occurs in average from 9 to 18% of women of reproductive age and includes a number of metabolic and endocrine dysfunctions [81]”
11) L220-222. Please add a reference supporting the phrase.
Response/corrections
There is both clinical and experimental data that treatment of patients (experimental animals) without signs of obesity and T2DM can cause significant changes in metabolic and hormonal parameters, which may be due to hyperactivation of MF-dependent signaling cascades. Accordingly, there are risks of using metformin in patients with normal metabolic and hormonal status, with normal glucose and insulin sensitivity, etc. In other words, it is necessary to carefully assess the functional and metabolic state of the patient to assess the feasibility and choice of a strategy for the use of metformin. At the same time, there are no data on the effect of metformin on the reproductive system in women and men with normal metabolic and hormonal parameters in the available literature, but possible risks must be taken into account. This is indicated by the phrases at the end of the first paragraph of the Section 3.2. In line with the commentary of the reviewer, we have added information on the available data from clinical and experimental studies on the effect of metformin on metabolic and hormonal parameters in nondiabetic individuals and healthy animals (page 5, lines 194-197).
«This possibility is supported by the data from clinical trials on metabolic changes, including an increase in fasting glucose clearance and endogenous glucose production [117,118], as well as changes in the microbiota in non-diabetic individuals [119], as well as data on metabolic and hormonal dysfunctions in normal rodents, for a long time receiving MF [120]».
12) In 3.2 it is not clear what spectrum of treatment of PCOS will be discussed. What PCOS treatment means?
Response/corrections
I agree that the title of section 3.2 does not accurately reflect its content and therefore it has been changed (new title «The use of metformin in PCOS women»). In addition, there is ambiguity in the use of the term "metformin therapy" for PCOS patients, and therefore the term used in section 3.2 has been deleted and the relevant phrases have been revised.
13) L302-316 refer to treatment of hyperglycemia more than specifically PCOS.
Response/corrections
I fully agree with the Reviewer that this paragraph is mainly devoted to the metabolic-enhancing effects of GLP-1 analogues in PCOS and the potentiation of these effects in the presence of metformin. Nevertheless, the improvement of metabolic effects, and, first of all, the weakening of HA and insulin resistance (hyperinsulinemia) are the molecular mechanisms of GLP-1 analogues-induced improvement of reproductive functions in PCOS patients. This information was missing in the paragraph and was added with explanations and supporting bibliographic references (page 7, lines 289-295):
“Weight loss and decrease in hyperglycemia and IR, which are induced by treatment of obese PCOS patients with GLP-1 receptor agonists, lead to a decrease in HA [167-169] and an improvement in menstrual frequency [167, 169]. Liraglutide, an analogue of GLP-1, normalizes the menstrual cycle and fertility in women with HAIR-AN syndrome, which is due to a decrease in the levels of androgens and insulin [170]. Consequently, in PCOS patients, MF-induced potentiation of the metabolic-improving effects of GLP-1 agonists may also increase their restorative effects on the menstrual cycle and fertility”.
14) L336-337 please rewrite the sentence, it is complex to understand.
Response/corrections
In accordance with the reviewer's comment, the sentence (lines 315-317) and the following have been rewritten for better understanding:
“In PCOS, the severity of IR is positively correlated with the severity of HA and dysregulations of the ovulatory cycle. The PCOS women with oligomenorrhea and without HA usually do not have IR, while the PCOS women with oligomenorrhea and HA often show significant signs of IR [182]”.
15) L349 Please correct Folluicle-stimulating Hormone.
Response/corrections
The typo has been corrected: “follicle-stimulating hormone”
16) L364-383 lack of references, and some sentences require bibliographical support (See the next point)
Response/corrections
This paragraph is not part of the main text, but a detailed description of the Figure 2. Unfortunately, the format of this description did not allow distinguishing it from the main text, and I apologize for the formatting inaccuracies. The formatting of this description has now been changed, and at the end, a phrase has been added stating that all additional details, including bibliographic references, are provided in the text in the Section 3.4.
17) L 364 Why it is stated that IR and androgen excess are the main pathogenic factors of PCOS? This is a complex affirmation due the uncertain PCOS pathogenesis. I agree that most of PCOS patients have IR and HA, but it is not clear that both factors are the cause of PCOS.
Response/corrections
I fully agree with the reviewer that at present, due to problems with the study of the genesis of PCOS, the use of the definition of "main pathogenic factors of PCOS" for the hyperinsulinemia (or IR) and hyperandrogenemia in PCOS is not fully justified, despite the wide prevalence of these hormonal dysfunctions in patients with PCOS. As a consequence, in accordance with the reviewer's comment, the corresponding statement was weakened (lines 369-371 in the description to the Figure 2):
“Hyperinsulinemia and HA are among the key pathogenetic factors in the development of PCOS, which is why, their attenuation by MF is the most important mechanism for improving effect of this drug on ovarian function in PCOS women”
18) L239-245 and L559-566 show the same information. Please not being redundant.
Response/corrections
In accordance with the reviewer's comment, the text on page 6 (lines 240-242) has been changed, and part of the text on page 12 (lines 535-537 and below) has been shortened, which partially repeats the information on page 6. Thus, the reduced paragraph was transformed into a sentence:
“…months [245]. By reducing the ovarian sensitivity to FSH, MF prevents the OHSS, the most common complication of gonadotropin-stimulated induction of ovulation [135, 136, 138, 255, 256]”
19) It is a little bit confusing that author say that MF increases the sensitivity to Gonadotropins but decreases the effects of CREB mediated transcription, since both LH and FSH has a Gs/AMPc/PKA/CREB mechanism.
Response/corrections
I fully agree with the reviewer that the sometimes opposite effects of metformin on gonadotropin signaling cascades and on gonadotropin sensitivity require further research. The regulatory effects of gonadotropins (LH, FSH) on target cells (the follicular cells, and the Leydig and Sertoli cells) can be mediated through several different signaling cascades, including through a cAMP-dependent pathway that includes the gonadotropin receptor, Gs protein, adenylyl cyclase, PKA or the exchange factor Epac, CREB or other cAMP-regulated transcription factors. In metabolic and endocrine disorders, these cascades can be overactive (as, for example, in the ovaries with PCOS) or, on the contrary, weakened (as, for example, in the testes under conditions of diabetes and metabolic syndrome). Moreover, changes in them can have a pronounced cell-specific character. It seems that such a complex pattern of disorders predetermines the difference in the action of metformin on sensitivity to gonadotropins and steroidogenic activity. In the case of PCOS, acting at various levels (hypothalamic regulation and pituitary gonadotropin production, receptors for them on target cells in the ovaries and testicles, post-receptor stages of gonadotropin signal transmission and their intracellular targets), metformin reduces hyperandrogenism and the overstimulating effect of FSH on follicular cells while in the case of androgen deficiency and impaired spermatogenesis in men with diabetes and metabolic syndrome, its effect is opposite. In the latter case, metformin treatment ensures the normalization of steroidogenesis and spermatogenesis, including through an improvement in the response to gonadotropins. Accordingly, the effect of metformin on the activity of CREB-dependent expression also differs.
20) Regarding AMH (3.4.5) please clarify if MF effect on AMH is direct (For example through AMPK action) or depends on normalization of follicular development by other mechanism. With a little sentence at the end of the section is enough.
Response/corrections
Thank you very much for your comment and since the possible mechanism of action of metformin on AMH production is of great importance, an additional paragraph has been added to the final part of the Section 3.4 (see below, lines 626-640 + new references). It provides the most likely mechanism based on the normalization of androgen levels. At the same time, other mechanisms cannot be ruled out, since control over the production of AMH can be carried out through other cascades and with the participation of a large number of transcription factors. So it is possible that the normalizing effect of metformin on AMH production is due to the restoration of expression and post-translational modification of proteins as a result of preventing the activation of the reaction of unfolded proteins triggered by excess androgens. In other words, metformin can normalize the production of AMH at different levels, reducing the level of androgens and peptides of the insulin superfamily (insulin, IGF-1), as well as weakening the negative effect of excess androgens on granulosa cells.
“Based on the above results, as well as on the available data on the molecular mechanisms mediating the regulation of AMH production by granulosa cells [249, 293], it can be assumed that the main mechanism for the improving effect of MF on AMH levels in PCOS is the weakening of HA. Under normal conditions, the androgens produced by theca cells induce a decrease in AMH levels, which leads to inhibition of the antral follicle development and precedes ovulation. With prolonged exposure to high concentrations of androgens, which are comparable to those in PCOS, the response of granulosa cells to androgens is impaired, resulting in the absence of an androgen-induced fall in AMH levels and dysregulation of follicular development [293]. When PCOS patients are treated with MF, their androgen levels are normalized, and hyperinsulinemia, which is usually associated with HA, is reduced, which leads to the restoration of the granulosa cell response to androgens. Accordingly, the low efficacy of MF in reducing AMH levels and restoring follicular maturation in patients with PCOS may be due to initially mild HA and IR. It is impossible to exclude the direct effects of MF on the production of AMH by follicular cells, including through AMPK-dependent mechanisms, as well as through weakening the stress of the endoplasmic reticulum, stimulated by high concentrations of androgens [237]. However, this issue has not yet been studied.”
21) L685-685. Please be cautious with the affirmation that something …is the main pathogenic mechanism... It is better to say for example “oxidative stress and inflammation are essential/crucial/very important/fundamental in the pathogenesis of PCOS”
Response/corrections
Thank you very much for your comment. I agree that the impact of these factors should not be overestimated. Corresponding phrase has been changed (page 15, lines 669-670):
“Dyslipidemia, oxidative stress and inflammation, along with IR, are essential for the pathogenesis of PCOS.”
22) L732-744 lack o references and repeat some previous ideas. If it is a conclusion paragraph, I suggest only a simple 2-3 lines sentence with the conclusion of the section.
Response/corrections
Similar to Comment 16.
This paragraph is not part of the main text, but a detailed description of the Figure 3. Unfortunately, the format of this description did not allow distinguishing it from the main text, and I apologize for the formatting inaccuracies. The formatting of this description has now been changed, and at the end, a phrase has been added stating that all additional details, including bibliographic references, are provided in the text in the Section 3.5.
23) L783-796 What is the mechanism by which metformin regulate vascular growth factors?
Response/corrections
One of the main mechanisms of action of metformin on preeclampsia is the regulation of expression and the optimal ratio of pro-angiogenic factors (vascular endothelial growth factor-A, placental growth factor) and anti-angiogenic factors (placenta-produced soluble vascular endothelial growth factor receptor-1, soluble endoglin). Since the level and activity of anti-angiogenic factors increases during preeclampsia, their decrease is an approach for prevention of preeclampsia, and this effect is achieved when patients are treated with metformin and its combinations with other drugs. Of course, in addition to changing the expression of these factors, it can be assumed that metformin also affects their signaling cascades within the cell. Moreover, the positive effect of metformin treatment on the endothelial dysfunctions may be important, including the improvement of mitochondrial dynamics, and the normalization of the ratio of vasoconstriction and vasodilation factors, etc.
In accordance with the commentary of the reviewer and in connection with the additions on the molecular mechanisms of action of metformin on angiogenesis, the corresponding paragraph in section 4 (lines 766-779, page 17) was changed:
« As noted above, there is evidence that MF reduces preeclampsia in GDM, and this is based on the ability of MF to prevent endothelial dysfunctions by normalizing the levels of the anti- and pro-angiogenic factors and improving the mitochondrial energy and biogenesis [313, 356, 357]. MF decreases the production of anti-angiogenic factors, including soluble forms of receptors of angiogenic factors. Among them, there are the placenta-produced soluble vascular endothelial growth factor receptor-1 (VEGFR-1), also called soluble fms-like tyrosine kinase-1 (sFlt-1), which specifically binds to vascular endothelial growth factor-A (VEGF-A) and placental growth factor (PlGF), and the soluble endoglin, a soluble isomer of endoglin, which inhibits the specific binding of TGF-β1 to its receptor [313, 356-358]. Moreover, MF and its combination with other drugs (esomeprazole, sulfasalazine) lead to an increase in the expression of VEGF-A and placental growth factor (PlGF), both powerful activators of angiogenesis, and to a decrease in TNFα-induced expression of endothelin-1, a potent vasoconstrictor [356, 357]. In pregnant women, the increased activity of anti-angiogenic factors induces the systemic endothelial dysfunctions and vasospasm and provokes preeclampsia [358]»
The new references
Bridges, H.R.; Jones, A.J.; Pollak, M.N.; Hirst, J. Effects of metformin and other biguanides on oxidative phosphorylation in mitochondria. Biochem. J. 2014, 462, 475–487. doi: 10.1042/BJ20140620.
Brownfoot, F.C.; Hastie, R.; Hannan, N.J.; Cannon, P.; Nguyen, T.V.; Tuohey, L.; Cluver, C.; Tong, S.; Kaitu'u-Lino, T.J. Combining metformin and sulfasalazine additively reduces the secretion of antiangiogenic factors from the placenta: Implications for the treatment of preeclampsia. Placenta. 2020, 95, 78-83. doi: 10.1016/j.placenta.2020.04.010.
Bryrup, T.; Thomsen, C.W.; Kern, T.; Allin, K.H.; Brandslund, I.; Jørgensen, N.R.; Vestergaard, H.; Hansen, T.; Hansen, T.H.; Pedersen, O.; Nielsen, T. Metformin-induced changes of the gut microbiota in healthy young men: results of a non-blinded, one-armed intervention study. Diabetologia. 2019, 62, 1024-1035. doi: 10.1007/s00125-019-4848-7.
Cena, H.; Chiovato, L.; Nappi, R.E. Obesity, Polycystic Ovary Syndrome, and Infertility: A New Avenue for GLP-1 Receptor Agonists. J. Clin. Endocrinol. Metab. 2020, 105, e2695–709. doi: 10.1210/clinem/dgaa285.
Choi, Y.K.; Park, K.G. Metabolic roles of AMPK and metformin in cancer cells. Mol. Cells. 2013, 36, 279-87. doi: 10.1007/s10059-013-0169-8.
Cioce, M.; Pulito, C.; Strano, S.; Blandino, G.; Fazio, V.M. Metformin: Metabolic Rewiring Faces Tumor Heterogeneity. Cells. 2020, 9, 2439. doi: 10.3390/cells9112439.
Davies, S.P.; Helps, N.R.; Cohen, P.T.; Hardie, D.G. 5'-AMP inhibits dephosphorylation, as well as promoting phosphorylation, of the AMP-activated protein kinase. Studies using bacterially expressed human protein phosphatase-2C alpha and native bovine protein phosphatase-2AC. FEBS Lett. 1995, 377, 421-425. doi: 10.1016/0014-5793(95)01368-7. PMID: 8549768.
Derkach, K.V.; Kuznetsova, L.A.; Sharova, T.S.; Ignat’eva, P.A.; Bondareva, V.M.; Shpakov, A.O. The effect of prolonged metformin treatment on the activity of the adenylate cyclase system and NO-synthase in the brain and myocardium of obese rats. Cell Tissue Biol. 2015, 9, 385–394. doi: 10.1134/S1990519X1505003X.
Dilaver, N.; Pellatt, L.; Jameson, E.; Ogunjimi, M.; Bano, G.; Homburg, R; Mason, H.; Rice, S. The regulation and signalling of anti-Müllerian hormone in human granulosa cells: relevance to polycystic ovary syndrome. Hum. Reprod. 2019, 34, 2467-2479. doi: 10.1093/humrep/dez214.
El-Mir, M.Y.; Nogueira, V.; Fontaine, E.; Averet, N.; Rigoulet, M.; Leverve, X. Dimethylbiguanide inhibits cell respiration via an indirect effect targeted on the respiratory chain complex I. J. Biol. Chem. 2000, 275, 223–228. doi: 10.1074/jbc.275.1.223.
Gao, F.; Chen, J.; Zhu, H. A potential strategy for treating atherosclerosis: improving endothelial function via AMP-activated protein kinase. Sci. China Life Sci. 2018, 61, 1024-1029. doi: 10.1007/s11427-017-9285-1.
Gormsen, L.C.; Søndergaard, E.; Christensen, N.L.; Brøsen, K.; Jessen, N.; Nielsen, S. Metformin increases endogenous glucose production in non-diabetic individuals and individuals with recent-onset type 2 diabetes. Diabetologia. 2019, 62, 1251–1256. doi: 10.1007/s00125-019-4872-7.
Hardie, D.G. AMPK: a key regulator of energy balance in the single cell and the whole organism. Int. J. Obes. (Lond). 2008, 32, S7-S12. doi: 10.1038/ijo.2008.116. PMID: 18719601.
Hardie, DG. Keeping the home fires burning: AMP-activated protein kinase. J. R. Soc. Interface. 2018, 15, 20170774. doi: 10.1098/rsif.2017.0774.
He, L. Metformin and Systemic Metabolism. Trends Pharmacol Sci. 2020, 41, 868-881. doi: 10.1016/j.tips.2020.09.001.
He, L.; Sabet, A.; Djedjos, S.; Miller, R.; Sun, X.; Hussain, M.A.; Radovick, S.; Wondisford, F.E. Metformin and insulin suppress hepatic gluconeogenesis through phosphorylation of CREB binding protein. Cell. 2009, 137, 635-646. doi: 10.1016/j.cell.2009.03.016.
Kaitu'u-Lino, T.J.; Brownfoot, F.C.; Beard, S.; Cannon, P.; Hastie, R.; Nguyen, T.V.; Binder, N.K.; Tong, S.; Hannan, N.J. Combining metformin and esomeprazole is additive in reducing sFlt-1 secretion and decreasing endothelial dysfunction - implications for treating preeclampsia. PLoS One. 2018, 13, e0188845. doi: 10.1371/journal.pone.0188845.
Karnewar, S.; Neeli, P.K.; Panuganti, D.; Kotagiri, S.; Mallappa, S.; Jain, N.; Jerald, M.K.; Kotamraju, S. Metformin regulates mitochondrial biogenesis and senescence through AMPK mediated H3K79 methylation: Relevance in age-associated vascular dysfunction. Biochim. Biophys. Acta Mol. Basis Dis. 2018, 1864, 1115-1128. doi: 10.1016/j.bbadis.2018.01.018.
Kim, J.; Yang, G.; Kim, Y.; Kim, J.; Ha, J. AMPK activators: mechanisms of action and physiological activities. Exp. Mol. Med. 2016, 48, e224. doi: 10.1038/emm.2016.16. PMID: 27034026; PMCID: PMC4855276.
Lamos, E.M.; Malek, R.; Davis, S.N. GLP-1 receptor agonists in the treatment of polycystic ovary syndrome. Expert. Rev. Clin. Pharmacol. 2017, 10, 401-408. doi: 10.1080/17512433.2017.1292125.
Lee, N.; Hebert, M.F.; Wagner, D.J.; Easterling, T.R.; Liang, C.J.; Rice, K.; Wang, J. Organic Cation Transporter 3 Facilitates Fetal Exposure to Metformin during Pregnancy. Mol. Pharmacol. 2018, 94, 1125-1131. doi: 10.1124/mol.118.112482.
Livadas, S.; Androulakis, I.; Angelopoulos, N.; Lytras, A.; Papagiannopoulos, F.; Kassi, G. Liraglutide administration improves hormonal/metabolic profile and reproductive features in women with HAIR-AN syndrome. Endocrinol. Diabetes Metab. Case Rep. 2020, 2020, 19-0150. doi: 10.1530/EDM-19-0150.
Lizcano, J.M.; Göransson, O.; Toth, R.; Deak, M.; Morrice, N.A.; Boudeau, J.; Hawley, S.A.; Udd, L.; Mäkelä, T.P.; Hardie, D.G.; Alessi, D.R. LKB1 is a master kinase that activates 13 kinases of the AMPK subfamily, including MARK/PAR-1. EMBO J. 2004, 23, 833-843. doi: 10.1038/sj.emboj.7600110.
Lyons C.L.; Roche. H.M. Nutritional Modulation of AMPK-Impact upon Metabolic-Inflammation. Int. J. Mol. Sci. 2018, 19, 3092. doi: 10.3390/ijms19103092.
McCreight, L.J.; Mari, A.; Coppin, L.; Jackson, N.; Umpleby, A.M.; Pearson, E.R. Metformin increases fasting glucose clearance and endogenous glucose production in non-diabetic individuals. Diabetologia. 2020, 63, 444–447. doi: 10.1007/s00125-019-05042-1.
Momcilovic, M.; Hong, S.P.; Carlson, M. Mammalian TAK1 activates Snf1 protein kinase in yeast and phosphorylates AMP-activated protein kinase in vitro. J. Biol. Chem. 2006, 281, 25336-25343. doi: 10.1074/jbc.M604399200.
Motoshima, H.; Goldstein, B.J.; Igata, M.; Araki, E. AMPK and cell proliferation--AMPK as a therapeutic target for atherosclerosis and cancer. J. Physiol. 2006, 574, 63-71. doi: 10.1113/jphysiol.2006.108324.
Ouyang, J.; Parakhia, R.A.; Ochs, R.S. Metformin activates AMP kinase through inhibition of AMP deaminase. J. Biol. Chem. 2011, 286, 1-11. doi: 10.1074/jbc.M110.121806.
Rattan, R.; Giri, S.; Hartmann, L.C.; Shridhar, V. Metformin attenuates ovarian cancer cell growth in an AMP-kinase dispensable manner. J. Cell. Mol. Med. 2011, 15, 166-178. doi: 10.1111/j.1582-4934.2009.00954.x. PMID: 19874425; PMCID: PMC3822503.
Sliwinska, A.; Drzewoski, J. Molecular action of metformin in hepatocytes: an updated insight. Curr. Diabetes Rev. 2015, 11, 175-181. doi: 10.2174/1573399811666150325233108.
Suter, M.; Riek, U.; Tuerk, R.; Schlattner, U.; Wallimann, T.; Neumann, D. Dissecting the role of 5'-AMP for allosteric stimulation, activation, and deactivation of AMP-activated protein kinase. J. Biol. Chem. 2006, 281, 32207-32216. doi: 10.1074/jbc.M606357200.
Tzotzas, T.; Karras, S.N.; Katsiki, N. Glucagon-Like Peptide-1 (GLP-1) Receptor Agonists in the Treatment of Obese Women with Polycystic Ovary Syndrome. Curr. Vasc. Pharmacol. 2017, 15, 218-229. doi: 10.2174/1570161114666161221115324.
Viollet, B.; Foretz, M. Revisiting the mechanisms of metformin action in the liver. Ann. Endocrinol. (Paris). 2013, 74, 123-129. doi: 10.1016/j.ando.2013.03.006.

Reviewer 2 Report
This is a long review summarising the published evidence on the mode of Metformin action in human disease with a specific emphasis on fertility in women and man.
It is pleasing to see that this review references most uptodate research and clinical trials. However, the overall length is excessive, therefore I would suggest more focus on the topic as indicated in the title and remove25% of the review which is not directly related. The pitfall of quoting other reviews and not original research creates a problem of repeating other authors biased reporting without scientific rigor. So can the author please focus on the original research, which will reduce the length of references.
Line 39 The paragraph starts …There is now a lot of evidence … At the end of this paragraph in the cluster of references 17-24 is provided. Out of 8 references none refers for the evidence on fertility.
It is well established and acknowledged by the author that metformin including other agents has tissue specific effects. The manuscript provides in some places large paragraphs without providing specific references (line 87 to line 114). Fig 1 does not specify where the data comes from to draw those mechanisms or of if this from another publication (permission is required).
Please reference line 221-222 statement.
Author Response
RESPONSE TO REVIEWER 2
Alexander O. Shpakov «Improvement effect of metformin on the female and male reproduction in endocrine pathologies and its mechanisms» (1016034)
COMMON SECTION OF RESPONSE TO REVIEWERS
I am very grateful to the Reviewers for a detailed analysis of the review article and for the comments made. I sincerely hope that the explanations and changes that I have made based on the comments and remarks of the Reviewers have improved the review article.
In accordance with the requirements of the Reviewers, I significantly revised the review article and made the following main changes and additions to it (in more details, the changes and corrections are presented below in the extended answers to the questions and comments of Reviewers 1 and 2).
- According to the comments of both reviewers, the Section 2 “Brief description of the targets and molecular mechanisms of action of metformin” has been extensively revised, restructured and supplemented with the necessary bibliographic references and explanations. In the process of revision of the Section 2, duplicate or redundant information was removed (see lines 51-155 in the manuscript).
- Unfortunately, due to inadequate text formatting (technical error), the descriptions (figure captions) for the Figures 2 and 3 began to be considered as parts of the main text. In this regard, these “redundant boxes” broke the logical presentation of the data and their discussion in the Sections 3.4 and 3.5, and began to be perceived as the conclusions to the corresponding sections. There were no bibliographic references in these “redundant boxes”, which led to the Comments 16 and 22 by Reviewer 1 and a general comment by Reviewer 2. These descriptions (figure captions) are now properly formatted and do not mix with the main text of the review.
- In accordance with the comments of the reviewers, the Section 6 "The effect of metformin on the offspring" has been removed from the review along with two tables. This is due to the fact that this section is not directly related to the general topic of the review, although it is of significant interest for medicine. I hope that removing the Section 6 improves the structure of the overview article and makes it easier to read. All sections have been modified as necessary due to deletion of the Section 6.
According to requirement of the Editor and Reviewers, all changes and corrections to the text are highlighted in yellow.
In conclusion, I would like to thank the Reviewers once again for the comments that helped improve the review article.
With best regards,
The Author
REVIEWER 2
Common comments
This is a long review summarizing the published evidence on the mode of Metformin action in human disease with a specific emphasis on fertility in women and man.
It is pleasing to see that this review references most update research and clinical trials. However, the overall length is excessive, therefore I would suggest more focus on the topic as indicated in the title and remove25% of the review which is not directly related. The pitfall of quoting other reviews and not original research creates a problem of repeating other authors biased reporting without scientific rigor. So can the author please focus on the original research, which will reduce the length of references?
Response/corrections
The Section 6 "The effect of metformin on the offspring" (about 20% of the review volume) has been removed from the review along with two tables. This is due to the fact that this section is not directly related to the general topic of the review, although it is of significant interest for reproduction, and also for improving the structure of the review. All sections have been modified as necessary due to deletion of the Section 6. In accordance with the reviewer's comment, original references were added in the process of finalizing various sections of the review, including those regarding the molecular mechanisms of action of metformin.
- Line 39 The paragraph starts …There is now a lot of evidence … At the end of this paragraph in the cluster of references 17-24 is provided. Out of 8 references none refers for the evidence on fertility.
Response/corrections
Unfortunately, the sentence including line 45 and below (the paragraph “There is now a lot of evidence…”), at the end of which there are references 17-24, is very long, and includes two different parts. The references 17-24 refer only to the second part of this sentence, devoted to the molecular mechanisms of action of metformin. To avoid ambiguity, the sentence has been divided into two sentences, and the references 17-24 refer only to the second sentence (lines 45-49, page 2 *):
“This review offers an overview of problems when utilizing MF therapy for the correction of reproductive dysfunctions in women and men and includes the analysis of possible mechanisms for positive effects of MF on reproduction. Review also includes only a brief description of the molecular mechanisms of MF action in target cells; these mechanisms are the focus of other review articles [17–24]”
* - Hereinafter, the Response / corrections column presents the new line and page numbering, while the Comments column retains the old numbering. The text from the new version of the article is given in quotation marks.
- It is well established and acknowledged by the author that metformin including other agents has tissue specific effects. The manuscript provides in some places large paragraphs without providing specific references (line 87 to line 114). Fig 1 does not specify where the data comes from to draw those mechanisms or of if this from another publication (permission is required).
Response/corrections
The Section 2 “Summary of cell targets and molecular mechanisms of action of metformin” has undergone a total revision, and the necessary bibliographic references and explanations are included in the text (see lines 51-155 in the manuscript).
Figure 1 is original and takes into account the results of the study of the mechanisms of metformin action, which are presented in the publications cited in the Section 2. In the Figure 1, in comparison with its earlier version, some adjustments were made related to the detailing of the relationship between metformin and its intracellular targets.
- Please reference line 221-222 statement.
Response/corrections
There is both clinical and experimental data that treatment of patients (experimental animals) without signs of obesity and T2DM can cause significant changes in metabolic and hormonal parameters, which may be due to hyperactivation of MF-dependent signaling cascades. Accordingly, there are risks of using metformin in patients with normal metabolic and hormonal status, with normal glucose and insulin sensitivity, etc. In other words, it is necessary to carefully assess the functional and metabolic state of the patient to assess the feasibility and choice of a strategy for the use of metformin. At the same time, there are no data on the effect of metformin on the reproductive system in women and men with normal metabolic and hormonal parameters in the available literature, but possible risks must be taken into account. This is indicated by the phrases at the end of the first paragraph of the Section 3.2. In line with the commentary of the reviewer, we have added information on the available data from clinical and experimental studies on the effect of metformin on metabolic and hormonal parameters in nondiabetic individuals and healthy animals (page 5, lines 194-197).
«This possibility is supported by the data from clinical trials on metabolic changes, including an increase in fasting glucose clearance and endogenous glucose production [117,118], as well as changes in the microbiota in non-diabetic individuals [119], as well as data on metabolic and hormonal dysfunctions in normal rodents, for a long time receiving MF [120]».
The new references
Bridges, H.R.; Jones, A.J.; Pollak, M.N.; Hirst, J. Effects of metformin and other biguanides on oxidative phosphorylation in mitochondria. Biochem. J. 2014, 462, 475–487. doi: 10.1042/BJ20140620.
Brownfoot, F.C.; Hastie, R.; Hannan, N.J.; Cannon, P.; Nguyen, T.V.; Tuohey, L.; Cluver, C.; Tong, S.; Kaitu'u-Lino, T.J. Combining metformin and sulfasalazine additively reduces the secretion of antiangiogenic factors from the placenta: Implications for the treatment of preeclampsia. Placenta. 2020, 95, 78-83. doi: 10.1016/j.placenta.2020.04.010.
Bryrup, T.; Thomsen, C.W.; Kern, T.; Allin, K.H.; Brandslund, I.; Jørgensen, N.R.; Vestergaard, H.; Hansen, T.; Hansen, T.H.; Pedersen, O.; Nielsen, T. Metformin-induced changes of the gut microbiota in healthy young men: results of a non-blinded, one-armed intervention study. Diabetologia. 2019, 62, 1024-1035. doi: 10.1007/s00125-019-4848-7.
Cena, H.; Chiovato, L.; Nappi, R.E. Obesity, Polycystic Ovary Syndrome, and Infertility: A New Avenue for GLP-1 Receptor Agonists. J. Clin. Endocrinol. Metab. 2020, 105, e2695–709. doi: 10.1210/clinem/dgaa285.
Choi, Y.K.; Park, K.G. Metabolic roles of AMPK and metformin in cancer cells. Mol. Cells. 2013, 36, 279-87. doi: 10.1007/s10059-013-0169-8.
Cioce, M.; Pulito, C.; Strano, S.; Blandino, G.; Fazio, V.M. Metformin: Metabolic Rewiring Faces Tumor Heterogeneity. Cells. 2020, 9, 2439. doi: 10.3390/cells9112439.
Davies, S.P.; Helps, N.R.; Cohen, P.T.; Hardie, D.G. 5'-AMP inhibits dephosphorylation, as well as promoting phosphorylation, of the AMP-activated protein kinase. Studies using bacterially expressed human protein phosphatase-2C alpha and native bovine protein phosphatase-2AC. FEBS Lett. 1995, 377, 421-425. doi: 10.1016/0014-5793(95)01368-7. PMID: 8549768.
Derkach, K.V.; Kuznetsova, L.A.; Sharova, T.S.; Ignat’eva, P.A.; Bondareva, V.M.; Shpakov, A.O. The effect of prolonged metformin treatment on the activity of the adenylate cyclase system and NO-synthase in the brain and myocardium of obese rats. Cell Tissue Biol. 2015, 9, 385–394. doi: 10.1134/S1990519X1505003X.
Dilaver, N.; Pellatt, L.; Jameson, E.; Ogunjimi, M.; Bano, G.; Homburg, R; Mason, H.; Rice, S. The regulation and signalling of anti-Müllerian hormone in human granulosa cells: relevance to polycystic ovary syndrome. Hum. Reprod. 2019, 34, 2467-2479. doi: 10.1093/humrep/dez214.
El-Mir, M.Y.; Nogueira, V.; Fontaine, E.; Averet, N.; Rigoulet, M.; Leverve, X. Dimethylbiguanide inhibits cell respiration via an indirect effect targeted on the respiratory chain complex I. J. Biol. Chem. 2000, 275, 223–228. doi: 10.1074/jbc.275.1.223.
Gao, F.; Chen, J.; Zhu, H. A potential strategy for treating atherosclerosis: improving endothelial function via AMP-activated protein kinase. Sci. China Life Sci. 2018, 61, 1024-1029. doi: 10.1007/s11427-017-9285-1.
Gormsen, L.C.; Søndergaard, E.; Christensen, N.L.; Brøsen, K.; Jessen, N.; Nielsen, S. Metformin increases endogenous glucose production in non-diabetic individuals and individuals with recent-onset type 2 diabetes. Diabetologia. 2019, 62, 1251–1256. doi: 10.1007/s00125-019-4872-7.
Hardie, D.G. AMPK: a key regulator of energy balance in the single cell and the whole organism. Int. J. Obes. (Lond). 2008, 32, S7-S12. doi: 10.1038/ijo.2008.116. PMID: 18719601.
Hardie, DG. Keeping the home fires burning: AMP-activated protein kinase. J. R. Soc. Interface. 2018, 15, 20170774. doi: 10.1098/rsif.2017.0774.
He, L. Metformin and Systemic Metabolism. Trends Pharmacol Sci. 2020, 41, 868-881. doi: 10.1016/j.tips.2020.09.001.
He, L.; Sabet, A.; Djedjos, S.; Miller, R.; Sun, X.; Hussain, M.A.; Radovick, S.; Wondisford, F.E. Metformin and insulin suppress hepatic gluconeogenesis through phosphorylation of CREB binding protein. Cell. 2009, 137, 635-646. doi: 10.1016/j.cell.2009.03.016.
Kaitu'u-Lino, T.J.; Brownfoot, F.C.; Beard, S.; Cannon, P.; Hastie, R.; Nguyen, T.V.; Binder, N.K.; Tong, S.; Hannan, N.J. Combining metformin and esomeprazole is additive in reducing sFlt-1 secretion and decreasing endothelial dysfunction - implications for treating preeclampsia. PLoS One. 2018, 13, e0188845. doi: 10.1371/journal.pone.0188845.
Karnewar, S.; Neeli, P.K.; Panuganti, D.; Kotagiri, S.; Mallappa, S.; Jain, N.; Jerald, M.K.; Kotamraju, S. Metformin regulates mitochondrial biogenesis and senescence through AMPK mediated H3K79 methylation: Relevance in age-associated vascular dysfunction. Biochim. Biophys. Acta Mol. Basis Dis. 2018, 1864, 1115-1128. doi: 10.1016/j.bbadis.2018.01.018.
Kim, J.; Yang, G.; Kim, Y.; Kim, J.; Ha, J. AMPK activators: mechanisms of action and physiological activities. Exp. Mol. Med. 2016, 48, e224. doi: 10.1038/emm.2016.16. PMID: 27034026; PMCID: PMC4855276.
Lamos, E.M.; Malek, R.; Davis, S.N. GLP-1 receptor agonists in the treatment of polycystic ovary syndrome. Expert. Rev. Clin. Pharmacol. 2017, 10, 401-408. doi: 10.1080/17512433.2017.1292125.
Lee, N.; Hebert, M.F.; Wagner, D.J.; Easterling, T.R.; Liang, C.J.; Rice, K.; Wang, J. Organic Cation Transporter 3 Facilitates Fetal Exposure to Metformin during Pregnancy. Mol. Pharmacol. 2018, 94, 1125-1131. doi: 10.1124/mol.118.112482.
Livadas, S.; Androulakis, I.; Angelopoulos, N.; Lytras, A.; Papagiannopoulos, F.; Kassi, G. Liraglutide administration improves hormonal/metabolic profile and reproductive features in women with HAIR-AN syndrome. Endocrinol. Diabetes Metab. Case Rep. 2020, 2020, 19-0150. doi: 10.1530/EDM-19-0150.
Lizcano, J.M.; Göransson, O.; Toth, R.; Deak, M.; Morrice, N.A.; Boudeau, J.; Hawley, S.A.; Udd, L.; Mäkelä, T.P.; Hardie, D.G.; Alessi, D.R. LKB1 is a master kinase that activates 13 kinases of the AMPK subfamily, including MARK/PAR-1. EMBO J. 2004, 23, 833-843. doi: 10.1038/sj.emboj.7600110.
Lyons C.L.; Roche. H.M. Nutritional Modulation of AMPK-Impact upon Metabolic-Inflammation. Int. J. Mol. Sci. 2018, 19, 3092. doi: 10.3390/ijms19103092.
McCreight, L.J.; Mari, A.; Coppin, L.; Jackson, N.; Umpleby, A.M.; Pearson, E.R. Metformin increases fasting glucose clearance and endogenous glucose production in non-diabetic individuals. Diabetologia. 2020, 63, 444–447. doi: 10.1007/s00125-019-05042-1.
Momcilovic, M.; Hong, S.P.; Carlson, M. Mammalian TAK1 activates Snf1 protein kinase in yeast and phosphorylates AMP-activated protein kinase in vitro. J. Biol. Chem. 2006, 281, 25336-25343. doi: 10.1074/jbc.M604399200.
Motoshima, H.; Goldstein, B.J.; Igata, M.; Araki, E. AMPK and cell proliferation--AMPK as a therapeutic target for atherosclerosis and cancer. J. Physiol. 2006, 574, 63-71. doi: 10.1113/jphysiol.2006.108324.
Ouyang, J.; Parakhia, R.A.; Ochs, R.S. Metformin activates AMP kinase through inhibition of AMP deaminase. J. Biol. Chem. 2011, 286, 1-11. doi: 10.1074/jbc.M110.121806.
Rattan, R.; Giri, S.; Hartmann, L.C.; Shridhar, V. Metformin attenuates ovarian cancer cell growth in an AMP-kinase dispensable manner. J. Cell. Mol. Med. 2011, 15, 166-178. doi: 10.1111/j.1582-4934.2009.00954.x. PMID: 19874425; PMCID: PMC3822503.
Sliwinska, A.; Drzewoski, J. Molecular action of metformin in hepatocytes: an updated insight. Curr. Diabetes Rev. 2015, 11, 175-181. doi: 10.2174/1573399811666150325233108.
Suter, M.; Riek, U.; Tuerk, R.; Schlattner, U.; Wallimann, T.; Neumann, D. Dissecting the role of 5'-AMP for allosteric stimulation, activation, and deactivation of AMP-activated protein kinase. J. Biol. Chem. 2006, 281, 32207-32216. doi: 10.1074/jbc.M606357200.
Tzotzas, T.; Karras, S.N.; Katsiki, N. Glucagon-Like Peptide-1 (GLP-1) Receptor Agonists in the Treatment of Obese Women with Polycystic Ovary Syndrome. Curr. Vasc. Pharmacol. 2017, 15, 218-229. doi: 10.2174/1570161114666161221115324.
Viollet, B.; Foretz, M. Revisiting the mechanisms of metformin action in the liver. Ann. Endocrinol. (Paris). 2013, 74, 123-129. doi: 10.1016/j.ando.2013.03.006.

Round 2
Reviewer 2 Report
Happy with the revised version. Thank you